

# Retrieving Atmospheric Thermodynamic and Hydrometeor Profiles Using a Thermodynamic-Constrained Kalman Filter 1D-Var Framework Based on Ground-based Microwave Radiometer

Qi Zhang[1,5], Tian Meng Chen[2], Jian Ping Guo[2], Bin Deng[3], Min Shao[4]

[1]Key Open Laboratory of Intelligent Meteorological Observation Technology, China Meteorological Administration, Beijing, 100081, China
[2]State Key Laboratory of Severe Weather Meteorological Science and Technology, Chinese Academy of Meteorological Sciences, Beijing, 100081, China
[3]Jiangxi Weather Modification Center, Nanchang, 330000, China
[4]School of Environment, Nanjing Normal University, Nanjing, 210023, China
[5]Engineering Technology Research and Development Center, China Huayun Meteorological Technology Group Co., Ltd., Beijing 100081, China

*Correspondence to*: Prof. Dr. Jian Ping Guo (jpguo@cma.gov.cn)

**Abstract.** Ground-based microwave radiometers (GMWRs) provide continuous thermodynamic profiling but suffer from degraded accuracy under cloudy and precipitating conditions when using classical one-dimensional variational (1D-Var) retrievals. To address this, we develop a thermodynamic-constrained Kalman filter variational framework (TCKF1D-Var) that enforces moist-thermodynamic consistency through the use of virtual potential temperature as the control variable, employs a ratio-based cost function independent of prescribed background and observation error covariances, and integrates a diagnostic microphysics closure to represent liquid and ice water. Validation over 43 GMWR sites in North China, including seven with collocated radiosondes, shows that TCKF1D-Var systematically reduces temperature and humidity biases relative to ERA5 and 1D-Var, with the largest improvements above 2 km for temperature and below 5.5 km for humidity. Temperature root-mean-square errors remain comparable to ERA5 and lower than 1D-Var below 8.5 km, while humidity errors are improved near the surface though degraded in the mid-troposphere due to vertical-resolution mismatch and channel cross-talk. Evaluation against collocated EarthCARE cloud liquid water content profiles demonstrates that TCKF1D-Var yields the lowest biases and errors and best reproduces observed distributions, confirming the benefit of the microphysics constraint. Case analyses of short-duration heavy rainfall further show that TCKF1D-Var enhances precursor signals of convection, extending the effective lead time for early warning relative to ERA5 and substantially outperforming 1D-Var. These results highlight the value of embedding physical constraints and microphysical closure within GMWR retrievals, offering a practical pathway to improve continuous thermodynamic monitoring and support high-impact weather nowcasting.



## 1 Introduction

High-resolution thermodynamic and hydrometeor profiles are essential for both atmospheric research and operational weather forecasting (Wulfmeyer et al. 2015; Wagner et al., 2019; Hu et al., 2019). Rapid vertical changes in temperature and water vapor associated with synoptic features can initiate high-impact weather events, including squall lines (Löhnert and Maier, 2012; Geerts et al., 2017) and mesoscale convective systems (Teixeira et al., 2025). Moreover, long-term, high-resolution datasets of temperature and humidity profiles in the PBL can help reveal how anthropogenic influences, such as urbanization, alter the thermodynamic structure (Barrera-Verdejo et al., 2021; Turner and Löhnert, 2021). Recognizing this need, the China Meteorological Administration (CMA) launched the "Weak-Link Remediation Project" in 2021, deploying ground-based microwave radiometers (GMWRs) and other instrumemts at selected sites to continuously retrieve high-resolution thermodynamic and hydrometeor profiles throughout the troposphere fill the observational gap between sparse radiosonde launches and satellite overpasses. This makes the GMWRs particularly valuable for monitoring fast-changing atmospheric signals and supporting short-range weather prediction.

However, the performance of ground-based microwave radiometer (GMWR) retrievals strongly depends on atmospheric conditions. Under clear-sky conditions, GMWR observations generally provide reliable temperature and humidity profiles with reasonable accuracy compared to radiosondes, regardless of methodological differences (Weisz et al., 2013; Ebell et al., 2017; Adler et al., 2021; Li et al., 2021; Xu, 2024). In contrast, under non-clear-sky conditions, significant retrieval biases can arise. Clouds introduce additional scattering and emission, particularly liquid water clouds that affect the 22–31 GHz water vapor absorption band, leading to overestimation of humidity and distortion of the retrieved vertical distribution (Zhang et al., 2024; Viggiano et al., 2025). Likewise, precipitation causes strong attenuation and scattering in both water vapor and oxygen absorption channels, degrading the information content for temperature retrievals (Kummerow et al., 2002; Christofilakis et al., 2020). These effects reduce retrieval reliability, especially in the lower troposphere where cloud and precipitation impacts are strongest.

Consequently, although GMWRs remain indispensable for continuous thermodynamic profiling, their application under cloudy and precipitating conditions requires advanced retrieval frameworks to mitigate these limitations. In this study, we introduce a novel Kalman filter–based one-dimensional variational optimal estimation framework (TCKF1D-Var) that integrates a thermodynamic conservation–constrained cost function with a cloud microphysics parameterization scheme, which generates retrievals of temperature, humidity, and hydrometeor profiles from microwave radiometer observations with higher accuracy. A comprehensive comparison with classical one-dimensional variational (1D-Var) method demonstrates that the proposed approach substantially improves retrieval accuracy, highlighting both its methodological novelty and its effectiveness in producing high-quality atmospheric profile products.



The remainder of this paper is organized as follows. Section 2 describes the study sites, instruments, and datasets employed.
Section 3 presents the technical framework and implementation of the 1D-Var and TCKF1D-Var optimal estimation methods. Section 4 evaluates the accuracy of thermodynamic profiles retrieved by these three frameworks under both daytime and nighttime conditions, as well as under different weather scenarios. Finally, Section 5 provides a summary and discusses the implications of the results for future research.

## 2 Data

### 2.1 GMWR Observation

In the North China region, the GMWRs deployed at different sites originate from various manufacturers, yet their channel configurations are consistent. Each instrument is equipped with seven water vapor channels (22.240, 23.040, 23.840, 25.440, 26.240, 27.840, and 31.400 GHz) and seven oxygen channels (51.260, 52.280, 53.860, 54.940, 55.500, 56.660, and 58.000 GHz), dedicated to observing the vertical distribution of atmospheric water vapor and temperature, respectively. Figure 1 illustrates the spatial distribution of MWRs across North China: in total, 43 stations are equipped with MWRs under the supervision of the CMA, among which 7 stations are co-located with conventional radiosonde launches. Simulated brightness temperatures, calculated from radiosonde profiles at these 7 stations using the RTTOV-gb radiative transfer model (De Angelis et al., 2016; Cimini et al., 2019), were compared with MWR observations. The results show that MWR measurements agree well with radiosonde-based simulations under clear-sky and cloudy conditions, while larger discrepancies occur during fog and precipitation events (Figure 2). Therefore, in this study, all optimal estimation retrievals are restricted to clear-sky and cloudy conditions in order to ensure both retrieval reliability and retrieval applicability.







**Figure 1. Spatial distribution of ground-based microwave radiometers (MWRs) in North China. Yellow markers denote stations equipped only with MWRs, while red markers indicate stations where MWRs are co-located with radiosonde launches.**





**Figure 2. Square root of the error covariance matrices of brightness temperature differences between ground-based microwave radiometer (MWR) observations and radiosonde-based simulations for different weather conditions: (a) clear-sky conditions, (b) cloudy conditions, (c) fog conditions, and (d) precipitation conditions.**




### 2.2 Radiosonde


Two radiosondes are launched daily from stations 53463, 53772, 54218, 54340, 54511, 54727, and 57083 (red dots in Figure 1), typically around 23:15 and 11:15 UTC. These routine soundings are employed as reference ("truth") data for constructing observation error covariance matrices and for evaluating the retrieval accuracy. The radiosondes provide high-quality vertical profiles with well-documented instrumental characteristics: temperature is measured with a resolution of 0.1 K and

an accuracy of 0.5 K, relative humidity with a resolution of 1% and an accuracy of 5%, and pressure with a resolution of 0.1 hPa and an accuracy of 0.5 hPa (Yao et al., 2025). Such specifications ensure that the radiosonde observations are sufficiently accurate to serve as an independent benchmark against which the ground-based microwave radiometer retrievals can be objectively assessed.

### 2.3 Hydrometeor Profile

To evaluate the performance of the optimal estimation framework for hydrometeor profiling, it is both sufficient and necessary to employ the EarthCARE (Earth Cloud, Aerosol, and Radiation Explorer, Kimura et al., 2003; Donovan et al., 2013; Hélière et al., 2017) cloud retrieval product (CPR_CLD_2A, Mason et al., 2024; Imura et al., 2025; European Space Agency, 2025) as a reference. The active radar observations from EarthCARE provide vertically resolved cloud liquid and ice water content with high sensitivity to optically thick clouds, which ensures the sufficiency of this dataset as a benchmark

for validating the vertical structures. Given the lack of long-term, ground-based observations with comparable vertical resolution and global coverage, the use of CPR_CLD_2A product is also a necessary step to establish the reliability and applicability of the optimal estimation hydrometeor retrievals in a broader context.

### 2.4 Priori Profile

In this study, the a priori atmospheric profile at each GMWR station is derived from the ERA5 reanalysis (Hoffmann et al.,

2019; Hersbach et al., 2020; Bell et al., 2021) using a bilinear interpolation method. ERA5 is widely recognized as one of the most reliable global reanalysis products, providing high temporal (hourly) and spatial (0.25° × 0.25°) resolution fields as well as a consistent assimilation of a large variety of observations. These advantages make ERA5 an appropriate substitute for direct observations in regions or periods where in situ measurements are sparse, discontinuous, or completely unavailable. The use of ERA5 as background information ensures that the constructed priori profiles capture large-scale atmospheric

variability with high fidelity, while still allowing the GMWR observations to provide additional fine-scale constraints during the retrieval process.



## 3 Methods

### 3.1 1D-Var Framework

As a widely used approach to retrieve atmospheric state, 1D-Var (Figure 3a) has been used in thermodynamic profile
retrieval systems for both ground-based (Hewison and Gaffard, 2006; Martinet et al., 2017; Gamage et al., 2020), airbone
(Thelen et al., 2009; Bell et al., 2021), and spacebone instruments (Noh et al., 2021; Wang et al., 2024; Carminati 2022) by
minimizing the cost function (Rodgers, 2000) Eq. (1):

$$J_{(x)} = (x - x_0)^T B^{-1}(x - x_0) + (y - H_{(x)})^T R^{-1}(y - H_{(x)}) \,, \tag{1}$$

where $y$ is the GMWR brightness temperature observation at a given time; $R$ is the observation error covariance matrix; $B$
is the background error covariance matrix; $x_0$ is the priori profile; and $H_{(x)}$ is the observation-operator-simulated
brightness temperature corresponding to a given atmospheric state $x$. In this study, the RTTOV-gb is selected as the
observation operator $H$.

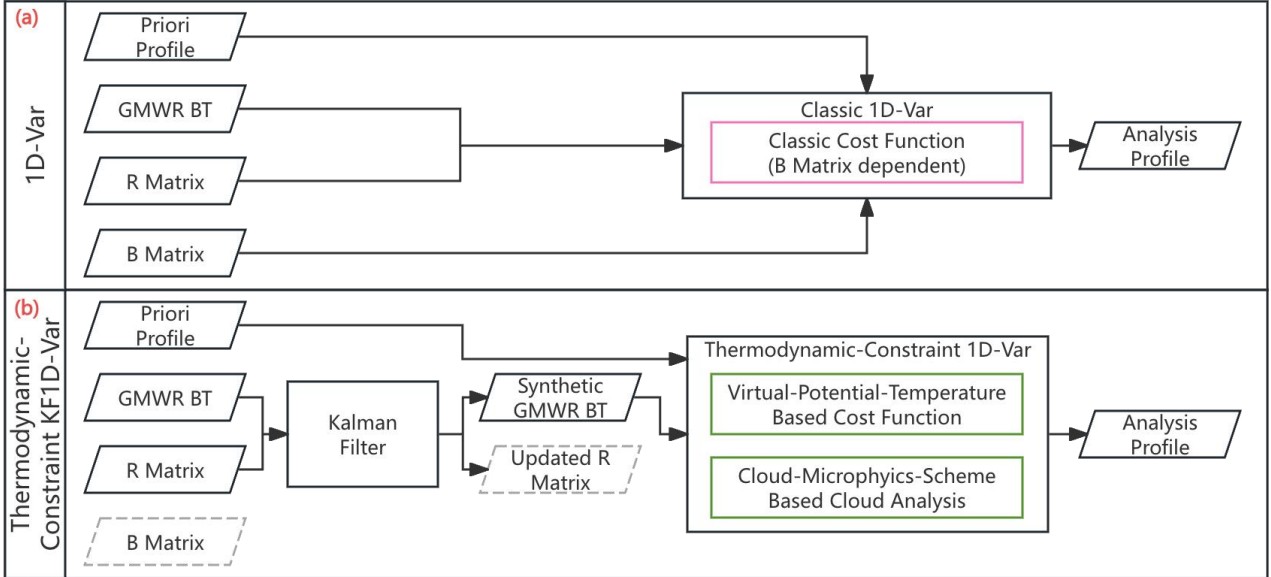

**Figure 3. The workflow of 1D-Var (a) and TCKF1D-Var (b) Framework.**

The accuracy of thermodynamic profiles retrieved with the 1D-Var framework depends not only on the measurement
precision of the instruments but also on the specification of the background and observation error covariance matrices, which
are typically estimated from long-term observational archives. While such climatologically based error covariances can





provide overall robust accuracy, their emphasis on representing the mean state reduces the ability of the retrieved optimal profiles to capture rapidly evolving weather phenomena, such as convective system initiation.

## 3.2 TCKF1D-Var Framework

The TCKF1D-Var framework (Figure 3b) inherits the Kalman filter's capability of reducing random measurement noise by generating synthetic observations (Welch and Bishop, 1995; Foth and Pospichal, 2017; Zhang et al., 2023, 2025) but replaces the cost function of the 1D-Var (Eq. 1) with a thermodynamic-constrained formulation in which the background and observation error covariance matrices are no longer required (Section 3.3.1). In addition, a microphysical hydrometeor analysis module (Section 3.3.2) is integrated into the 1D-Var framework to enhance the accuracy of the retrieved profiles under cloudy conditions.

### 3.2.1 Thermodynamic-Constrained Cost Function

Since virtual potential temperature ($\theta_v$) accounts for the influences of air pressure ($P$), temperature ($T$), water vapor ($q_v$), and hydrometeors ($q_l$ for cloud liquid water content, $q_i$ for cloud ice water content) in its calculation (Eq. (2) and (3)) and is conserved during moist adiabatic processes in the atmosphere (de Haan and van der Veen, 2014; Benjamin et al., 2021), it serves as an ideal control variable in the cost function. In addition, using this control variable not only allows for the adjustment of temperature and humidity profiles based on observations but also enables simultaneous modifications to pressure and hydrometeor profiles. More importantly, compared to classic 1D-Var (Section 3.1), this control variable ensures that the retrieved profiles satisfy thermodynamic equilibrium while achieving the mathematical optimum of the cost function.

$$\theta_{v(x)} = T(\frac{P_o}{P})^K (1 + 0.61 q_v - q_l - q_i) \,, \tag{2}$$

$$K = \frac{R_d}{C_p} \,, \tag{3}$$

where $R_d$ is the specific gas constant for dry air, and $C_p$ is the specific heat capacity at constant pressure for dry air.

To eliminate the influence of climatological background and observation error covariance matrices on the retrieved profiles, while ensuring that both the observation and control variable terms in the cost function are dimensionless, we replace the classic difference-based calculation in the cost function with a ratio-based formulation. Additionally, to enhance the contribution of the initial analysis increment to the cost function and reduce the number of iterations, unity (1, in this case) is subtracted from both the observation and control variable terms before squaring. The newly formulated cost function is expressed as in Eq. (4):



$$J_{(x)} = (\frac{H_{(x)}}{y} - 1)^2 + (\frac{\theta_{v(x)}}{\theta_{v(x_o)}} - 1)^2 , \tag{4}$$

**3.2.2 Microphysical Hydrometeor Analysis**

While RTTOV-gb incorporates the influence of cloud liquid water in brightness temperature calculations and provides the corresponding Jacobians, the inherently discontinuous vertical structure of clouds in the real atmosphere prevents liquid water profiles from being retrieved in a manner consistent with temperature and humidity profiles. Moreover, RTTOV-gb accounts only for liquid water, neglecting the impact of cloud ice. As a result, constructing the observational term of the cost

function solely with RTTOV-gb is insufficient to ensure its physical consistency and closure. Therefore, the inclusion of a cloud microphysics scheme in the cost function is essential to achieve a physically consistent and closed formulation. Considering the trade-off between computational efficiency and simulation accuracy, the WSM3 single-moment microphysics scheme (Hong et al., 2004; Que et al., 2016) is employed as the basis for the diagnostic representation of cloud liquid water and cloud ice profiles.

**3.3 Cost Function Minimization**

For the three frameworks described above, we employ the L-BFGS method (Limited-memory Broyden-Fletcher-Goldfarb-Shanno, Liu and Nocedal, 1989; Byrd et al., 1995) to obtain their optimal estimates. The rationale for choosing this method is as follows: (1) it has low memory requirements, making it suitable for high-dimensional optimization problems; (2) it does not explicitly store the Hessian matrix, but instead approximates it using the gradients and variable changes from the most

recent m steps, resulting in fast convergence. In this study, all three frameworks share the same parameter settings: a maximum of 1500 iterations, a cost function convergence tolerance of $3 \times 10^{-9}$, a gradient norm convergence tolerance of $1 \times 10^{-5}$, and a maximum of 20 line searches per iteration.

**4 Results**

**4.1 Thermodynamic Profile Evaluation**

**4.1.1 General Performance**

Figure 4 presents the validation results of TCKF1D-Var and 1D-Var, and the a priori profiles (ERA5) against radiosonde observations. As shown in Figure 4a, in terms of mean bias, the temperature profiles retrieved by TCKF1D-Var exhibit smaller average errors than the ERA5 a priori profiles, whereas the profiles derived from 1D-Var shows larger mean bias than the a priori. This indicates that TCKF1D-Var is capable of effectively correcting the systematic biases in the ERA5

temperature profiles, while the corrections achieved by 1D-Var are less evident. Moreover, the bias reduction provided by TCKF1D-Var is more pronounced in the free atmosphere (above 2000 m) than within the boundary layer (below 2000 m). In





terms of root mean square error (RMSE, Figure 4b), the random errors of the TCKF1D-Var temperature profiles are comparable to those of the ERA5 a priori below 8500 m above ground level, but become larger than the a priori above that level. Nevertheless, the overall random errors of TCKF1D-Var remain smaller than that from 1D-Var, highlighting that the
TCKF1D-Var framework, which incorporates virtual potential temperature conservation, is more suitable for tropospheric temperature profile retrievals compared to 1D-Var framework. For water vapor, the improvement in mean bias correction achieved by TCKF1D-Var is even more evident than for temperature (Figure 4c). Within 0–1000 m, the mean bias of TCKF1D-Var water vapor profiles is below 1.5 g/kg, whereas the ERA5 a priori bias exceeds 1.5 g/kg. Between 1000–5500 m, TCKF1D-Var reduces the mean bias to below 0.5 g/kg, while the ERA5 a priori bias remains between 0.5 and 1.5 g/kg.
In contrast, the water vapor profiles produced by 1D-Var show mean biases consistently larger than 0.5 g/kg below 9000 m. These results clearly demonstrate that, relative to 1D-Var, the TCKF1D-Var framework not only provides a more suitable approach for tropospheric temperature retrievals but also substantially reduces the mean bias of a priori water vapor profiles. Regarding RMSE (Figure 4d), the random errors of water vapor profiles retrieved with TCKF1D-Var are smaller than those of 1D-Var in the 0–1500 m range, but remain larger than the ERA5 a priori. In the 1500–5500 m layer, TCKF1D-Var
exhibits the largest RMSE among all products, while in the 5500–10000 m layer, its random errors are comparable to those of 1D-Var but still higher than those of the a priori.





**Figure 4. Validation of temperature and water vapor profiles retrieved by TCKF1D-Var (red), ERA5 a priori (cyan), and 1D-Var (blue) against radiosonde observations. For temperature, TCKF1D-Var reduces the mean bias (pannel a) relative to the a priori and yields smaller overall random errors (pannel b) than 1D-Var. For water vapor, TCKF1D-Var substantially decreases the mean bias (pannel c) compared to the a priori, particularly below 5500 m, while its random errors (pannel d) are smaller than 1D-Var near the surface but remain larger than the a priori aloft.**

**4.1.2 Day-Night Performance Difference**

Building on Section 4.1.1, we further separate the validation results of the four sets of profiles against radiosonde observations at 00:00 UTC (08:00 BJT) and 12:00 UTC (20:00 BJT) to examine the diurnal variability in the performance of TCKF1D-Var and 1D-Var, and the ERA5 a priori profiles. For temperature mean bias, the differences among the three products are mainly confined to the boundary layer. As shown in Figure 5a, at 00:00 UTC, the temperature profiles from TCKF1D-Var exhibit smaller systematic biases than the ERA5 a priori, indicating a more effective correction of the a priori bias, while the 1D-Var bias is larger. At 12:00 UTC (Figure 5e), both TCKF1D-Var and the a priori show smaller mean biase than 1D-Var. Although the difference between TCKF1D-Var and the a priori becomes less pronounced during daytime,





the TCKF1D-Var bias remains lower than that of the a priori. For RMSE, the 00:00 UTC results (Figure 5b) are generally consistent with those in Section 4.1.1. However, at night (Figure 5f), the random errors of the TCKF1D-Var temperature profiles increase substantially above 2000 m, shifting from being significantly smaller than 1D-Var during daytime to comparable levels. Regarding water vapor mean bias (Figures 5c and 5g), the diurnal validation results are largely consistent

with those in Section 4.1.1. The main difference appears below 500 m during nighttime, where, based on the 00:00 UTC radiosondes, TCKF1D-Var shows positive mean bias, while at night the mean bias becomes negative. Similarly, for RMSE (Figures 5d and 5h), the overall behavior resembles that in Section 4.1.1, with the only notable difference occurring above 5500 m during daytime: based on the 00:00 UTC radiosondes, 1D-Var produces larger random errors than TCKF1D-Var, whereas at night the two are nearly indistinguishable.






**Figure 5. Validation of temperature and water vapor profiles retrieved by TCKF1D-Var (red), ERA5 a priori (cyan), and 1D-Var (blue) against radiosonde observations at 00:00 UTC (a–d) and 12:00 UTC (e–h). Panels (a, e) show mean temperature bias, indicating that TCKF1D-Var provides the most effective correction of systematic errors relative to the a priori. Panels (b, f) present temperature RMSE, where TCKF1D-Var maintains smaller random errors than 1D-Var during daytime, but shows increased errors above 2000 m at night. Panels (c, g) show mean water vapor bias, with TCKF1D-Var substantially reducing systematic errors compared to the other methods, while differences between 1D-Var are most evident above 5500 m. Panels (d, h) depict water vapor RMSE, again showing improved performance of TCKF1D-Var near the surface, with daytime differences between 1D-Var diminishing at night.**





### 4.1.3 Accuracy under Different Weather Conditions

Under clear-sky and cloudy conditions (Figure 6a and e), the mean temperature errors are generally consistent with the results in Figure 4a, with TCKF1D-Var showing smaller biases than both ERA5 (as the background profiles) and 1D-Var. Under foggy (Figure 6i) and rainy (Figure 6n) conditions, TCKF1D-Var also exhibits reduced temperature errors below 5 km compared to ERA5 and 1D-Var, while above 5 km its performance is comparable to 1D-Var. In contrast, ERA5 shows similar errors to 1D-Var below 3 km but becomes less accurate above this level. Across all four weather regimes (clear-sky,

cloudy, foggy, and rainy), the root-mean-square errors (RMSEs) of the temperature profiles are consistent with Fig. 4b, i.e., TCKF1D-Var performs comparably to ERA5 but clearly outperforms 1D-Var. For water vapor, under clear-sky (Figure 6c) and cloudy (Figure 6g) conditions, the mean errors of TCKF1D-Var remain smaller than those of ERA5 and 1D-Var, consistent with the results reported earlier. Under foggy (Figure 6k) and rainy (Figure 6o) conditions, TCKF1D-Var again yields smaller mean errors than both ERA5 and 1D-Var, although ERA5 exhibits larger biases than 1D-Var within the

boundary layer (below ~500 m in foggy cases and below ~1000 m in rainy cases). In all four weather regimes, the RMSEs of water vapor profiles from TCKF1D-Var are comparable to those of ERA5 and consistently lower than those of 1D-Var.







**Figure 6.** Mean errors (a, e, i, n) and root-mean-square errors (b, f, j, o for temperature; c, g, k, p for water vapor) of retrieved profiles under four weather conditions: clear-sky, cloudy, foggy, and rainy. TCKF1D-Var consistently shows smaller temperature biases than ERA5 and 1D-Var below 5 km, with performance comparable to ERA5 above this level. ERA5 exhibits higher errors than 1D-Var above 3 km. For water vapor, TCKF1D-Var outperforms both ERA5 and 1D-Var across all regimes, while ERA5 displays larger boundary-layer errors (below ~500 m in foggy cases and below ~1000 m in rainy cases). In all conditions, the RMSEs of TCKF1D-Var are comparable to ERA5 and lower than 1D-Var.





### 255   4.1.4 Water Vapor RMSE Deficit Analysis

Despite the overall accuracy improvements achieved by the TCKF1D-Var framework, the retrieved water vapor profiles still exhibit systematic degradation in the middle troposphere (approximately 1.5–4.5 km) from the aspect of RMSE. A primary cause of this problem is the inherent coupling between temperature and humidity signals in the GMWR-measured brightness temperatures. While oxygen channels provide constraints on the temperature structure, their weighting functions peak at

vertical levels, calculated by PyRTlib (Larosa et al., 2024) using US Standard Atmosphere profile as background (NOAA, 1976), do not coincide with those of the water vapor channels (Figure 7). This vertical resolution mismatch leads to a partial leakage of temperature uncertainties into the humidity retrieval, thereby amplifying errors in this altitude range. The effect is particularly pronounced where temperature-sensitive channels show weak sensitivity, while humidity-sensitive channels remain reasonably responsive, leaving the retrieval under-constrained and more dependent on the completeness of the cost

function design. Such temperature–humidity coupling has been reported in earlier radiative transfer and retrieval studies (e.g., Hewison, 2007; Löhnert and Maier, 2012), underscoring the necessity of explicitly characterizing vertical resolution mismatches and accounting for cross-variable error propagation when developing retrieval strategies.







**Figure 7. Weighting functions of temperature-sensitive (oxygen) and humidity-sensitive (water vapor) channels calculated with**
**PyRTlib using the US Standard Atmosphere profile (NOAA, 1976) as background. The mismatch in vertical sensitivity between**
**oxygen and water vapor channels is highlighted.**

**4.2 Hydrometeor Profile Evaluation**

Previous evaluations of temperature and humidity have demonstrated the higher accuracy of the TCKF1D-Var profiles. In

this section, we further validate the three retrieval products against the EarthCARE cloud liquid water content (CLWC)

observations. The EarthCARE profiles were collocated in time and space by applying the following criterion: if an

EarthCARE observation occurred within ±15 min of the profile validation time and within a 15 km radius of the station, the

corresponding EarthCARE CLWC profile was used as the reference truth. Table 1 summarizes the number of collocated

cases and profiles available in July 2025.

**Table 1. Summary of collocated EarthCARE cloud liquid water content (CLWC) profiles and corresponding retrieval cases used**
**for validation in July 2025.**



| WMO Station ID | Latitude (degree north) | Longitude (degree east) | Evaluation Profile Amount |
|---|---|---|---|
| 53487 | 40.08 | 113.42 | 2 |
| 53662 | 38.72 | 111.58 | 2 |
| 53982 | 35.23 | 113.27 | 2 |
| 54398 | 40.13 | 116.62 | 2 |
| 54406 | 40.45 | 115.97 | 2 |
| 54412 | 40.73 | 116.63 | 2 |
| 54433 | 39.95 | 116.50 | 2 |
| 54505 | 39.94 | 116.10 | 2 |
| 54514 | 39.87 | 116.25 | 2 |
| 54594 | 39.72 | 116.35 | 4 |
| 54751 | 37.94 | 120.73 | 2 |
| 57171 | 33.77 | 113.12 | 4 |
| 53588 | 38.95 | 113.52 | 2 |
| 53673 | 38.73 | 112.72 | 4 |
| 53760 | 37.88 | 111.23 | 4 |
| 53959 | 35.11 | 111.07 | 4 |
| 54399 | 39.98 | 116.28 | 2 |
| 54410 | 40.60 | 116.13 | 2 |
| 54419 | 40.37 | 116.63 | 2 |
| 54424 | 40.17 | 117.12 | 4 |
| 54501 | 39.98 | 115.69 | 3 |
| 54511 | 39.80 | 116.47 | 2 |
| 54525 | 39.73 | 117.28 | 8 |
| 54727 | 36.68 | 117.55 | 2 |
| 58025 | 34.57 | 117.73 | 4 |

As shown in Figure 8, the mean errors indicate that the 1D-Var retrieval underestimates LWC by $10^1 - 10^2$ mg·m$^{-3}$ relative to EarthCARE in the 0–6 km layer, while ERA5 shows a smaller underestimation of about $10^0 - 10^1$ mg·m$^{-3}$. In contrast, TCKF1D-Var exhibits the smallest bias, with deviations consistently within $10^0$ mg·m$^{-3}$ throughout the 0–6 km range. In terms of RMSE, TCKF1D-Var maintains values below $10^2$ mg·m$^{-3}$ across 0–6 km, whereas ERA5 exceeds $10^2$ mg·m$^{-3}$ except below 1 km, and 1D-Var shows the largest errors, remaining in the range of $10^2$–$10^3$ mg·m$^{-3}$.








**Figure 8. Figure 8. Mean error and root-mean-square error (RMSE) of cloud liquid water content (CLWC) profiles from 1D-Var, ERA5, and TCKF1D-Var relative to EarthCARE observations in the 0 – 6 km layer. The 1D-Var retrievals show the largest underestimation ($10^1$–$10^2$ mg·m$^{-3}$) and highest RMSE ($10^2$–$10^3$ mg·m$^{-3}$), ERA5 exhibits smaller biases ($10^0$–$10^1$ mg·m$^{-3}$) but RMSEs above $10^2$ mg·m$^{-3}$ except below 1 km, while TCKF1D-Var achieves the smallest deviations, with mean errors within $10^0$ mg·m$^{-3}$ and RMSEs consistently below $10^2$ mg·m$^{-3}$.**

The frequency distribution histogram (Figure 9) of CLWC further confirm these findings. TCKF1D-Var most closely matches the EarthCARE distribution below 136 mg·m$^{-3}$ particularly in the 36–136 mg·m$^{-3}$ interval where the agreement is strongest. In the 0–36 mg·m$^{-3}$ range, TCKF1D-Var differs from EarthCARE by an order of $10^1$, while ERA5 and 1D-Var show discrepancies exceeding $10^2$. For the 136–170 mg·m$^{-3}$ interval, the differences of TCKF1D-Var relative to EarthCARE are comparable to, or slightly larger than, those of ERA5 and 1D-Var.



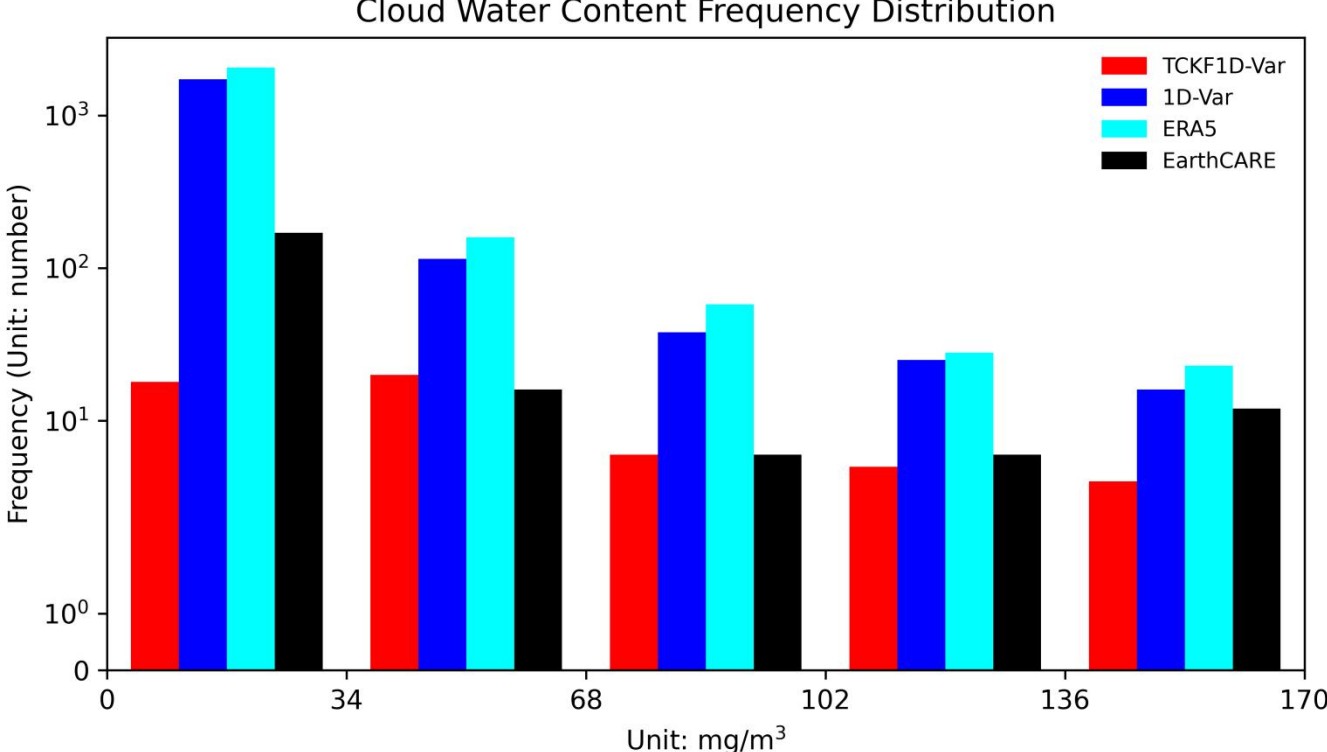

**Figure 9. Frequency distribution histograms of cloud liquid water content (CLWC) from 1D-Var, ERA5, and TCKF1D-Var**
**compared with EarthCARE observations. TCKF1D-Var agrees most closely with EarthCARE below 136 mg·m⁻³, particularly in**
**the 36−136 mg·m⁻³ interval. In the 0−36 mg·m⁻³ range, TCKF1D-Var differs from EarthCARE by about one order of magnitude**
**(10¹), while ERA5 and 1D-Var show larger discrepancies exceeding 10². For 136−170 mg·m⁻³, the deviations of TCKF1D-Var**
**relative to EarthCARE are comparable to, or slightly greater than, those of ERA5 and 1D-Var.**

**4.3 Extreme Precipitation Event Early-warning Capability Demonstration**

The preceding results demonstrate that the thermodynamic profiles retrieved from TCKF1D-Var framework exhibit smaller
mean biases than ERA5, with root-mean-square errors lower than those from 1D-Var and comparable to ERA5, while the
retrieved cloud liquid water content shows a higher degree of consistency with EarthCARE observations compared to both
1D-Var and ERA5. To further confirm that these improvements in retrieval accuracy translate into practical benefits, we
investigate their implications for the early identification of extreme precipitation signals. Using the criterion of hourly
accumulated precipitation exceeding 10 mm to define heavy rainfall events (World Meteorological Organization, 2007), we
identified eight short-duration extereme precipitation cases in July 2025 at stations equipped with GMWRs (Table 2).
Following the approach proposed by Taylor et al. (2007) and Garcia-Carreras et al. (2010), we adopt the temporal moving
anomaly of virtual potential temperature as an early-warning indicator. Using the selected precipitation cases, we analyze the
time series derived from different profile products and evaluate its relationship with the onset of heavy rainfall events. The
temporal deviation of virtual potential temperature is calculated as follows:





$$\theta_v^{anomaly} = \theta_v^{T0} - \frac{\Sigma_{t=T_0-Window\ Size}^{T_0} \theta_v^t}{Window\ Size}, \tag{5}$$

where $\theta_v^{anomaly}$ is the temporal moving anomaly of virtual potential temperature, $\theta_v^{T0}$ is the  virtual potential temperature at

observation time $T_0$, $\theta_v^t$ is the  virtual potential temperature at time $t$ within the time window of $Window\ Size$.

**Table 2. List of short-duration extreme precipitation cases (hourly accumulation >10 mm) observed in July 2025 at stations**
**equipped with GMWRs.**

| WMO Station ID | Start Date and Time (UTC) | End Date and Time (UTC) | Maximum Precipitation (mm/hr) |
|---|---|---|---|
| 54727 | 2025-07-01 19:00:00 | 2025-07-01 21:00:00 | 13.7 |
| 57083 | 2025-07-01 17:00:00 | 2025-07-01 17:00:00 | 11.0 |
| 53772 | 2025-07-07 07:00:00 | 2025-07-07 07:00:00 | 15.5 |
| 53673 | 2025-07-09 17:00:00 | 2025-07-09 19:00:00 | 17.0 |
| 54727 | 2025-07-23 04:00:00 | 2025-07-23 04:00:00 | 14.5 |
| 53463 | 2025-07-25 07:00:00 | 2025-07-25 08:00:00 | 12.8 |
| 54511 | 2025-07-27 17:00:00 | 2025-07-27 19:00:00 | 10.4 |
| 54511 | 2025-07-28 20:00:00 | 2025-07-28 20:00:00 | 12.2 |

Figure 10 presents the case-averaged time–height evolution of the virtual potential temperature anomaly derived from ERA5 (Figure 10a, d, and g), TCKF1D-Var (Figure 10b, e, and h), and 1D-Var (Figure 10c, f, and i) under different temporal averaging windows: 9.0-hour (Figure 10a, b, and c), 10.5-hour (Figure 10d, e, and f), and 12.0-hour (Figure 10g, h, and i),
spanning from 11 hours prior to the onset of precipitation to the time of rainfall occurrence. From the ERA5 profiles, a distinct signal emerges below 400 m, where the anomaly changes from positive to negative during the 11 h preceding precipitation, accompanied by the intrusion of a warm anomaly tongue between 400 and 1100 m. Taking the transition from +0.25 K to −0.25 K as the early-warning threshold, ERA5 allows the identification of heavy rainfall 6–7 hour in advance. When using the criterion of the anomaly dropping below −0.75 K as a secondary trigger, ERA5 can reconfirm the
occurrence of heavy rainfall 4–5 hour ahead. The TCKF1D-Var retrievals reproduce these key precursory features, namely the positive-to-negative anomaly transition and the warm tongue intrusion, with an even stronger signal compared to ERA5. Based on the +0.25 K to −0.25 K transition, TCKF1D-Var indicates the potential onset of heavy rainfall 7.5–8.0 hour in advance, while the secondary threshold of −0.75 K enables a reconfirmation 4.0–4.5 hour before the event. By contrast, the 1D-Var profiles fail to capture the anomaly transition and warm tongue intrusion in the pre-precipitation stage, and the +0.25
K to −0.25 K transition cannot be identified as a reliable precursor. Moreover, when adopting the −0.75 K criterion, the 1D-Var time–height series exhibits two spurious anomaly layers, leading to an increased false-alarm rate. In summary, the TCKF1D-Var framework enhances the representation of vertical atmospheric structures relevant to heavy rainfall initiation,





such as the temporal evolution of virtual potential temperature anomalies and warm tongue intrusions, thereby providing a

slightly longer lead time for early-warning signals compared to ERA5, whereas such improvements are absent in the 1D-Var

results.



**Figure 10. Case-averaged time–height evolution of virtual potential temperature anomalies derived from ERA5 (a, d, g), TCKF1D-Var (b, e, h), and 1D-Var (c, f, i) under different temporal averaging windows of 9.0-hour (a–c), 10.5-hour (d–f), and 12.0-hour (g–i), spanning from 11 h before to the onset of precipitation. ERA5 and TCKF1D-Var reveal the positive-to-negative anomaly**
**transition below 400 m and the warm anomaly tongue intrusion between 400–1100 m, serving as precursors of heavy rainfall. Compared to ERA5, TCKF1D-Var provides stronger signals and longer lead times (7.5–8.0 h), whereas 1D-Var fails to capture these features and exhibits spurious anomaly layers under the −0.75 K criterion.**

Using the same methodology, we recalculated the time–height evolution of the virtual potential temperature anomaly with a

reduced temporal averaging window. As shown in Figure 11, the gradients of the anomaly variations become weaker

compared to those in Figure 10, owing to the shorter averaging window. Nevertheless, both ERA5 (Figure 11a, d, and g) and

TCKF1D-Var (Figure 11b, e, and h) profiles still exhibit the characteristic transition of the anomaly from positive to



negative about 7–8 h prior to rainfall onset. Although the warm anomaly tongue intrusion remains detectable in both products, its intensity is reduced. When adopting −0.75 K as the early-warning threshold, the signal becomes indistinct under the 4.5-hour averaging window, whereas it is enhanced and temporally stabilized within about 2 hours of the precipitation onset when using 6.0-hour and 7.5-hour windows. Consistent with the previous findings, the 1D-Var (Figure 11c, f, and i) profiles fail to extract effective early-warning signals for heavy rainfall.

**Figure 11. Case-averaged time–height evolution of virtual potential temperature anomalies derived from ERA5 (a, d, g), TCKF1D-Var (b, e, h), and 1D-Var (c, f, i) under reduced temporal averaging windows of 4.5-hour (a–c), 6.0-hour (d–f), and 7.5-hour (g–i). Compared to Fig. 10, anomaly gradients weaken with shorter windows; however, ERA5 and TCKF1D-Var still capture the positive-to-negative transition ~7–8 hours before rainfall onset and the warm anomaly tongue intrusion, albeit with reduced intensity. The −0.75 K early-warning signal is indistinct for the 4.5-hour window but becomes clearer and more temporally stable (~2 hours offset) for the 6.0 h and 7.5 h windows, while 1D-Var fails to provide effective precursors.**





## 5 Summary and Concluding Remarks

This study introduces and evaluates TCKF1D-Var, a thermodynamic-constrained Kalman-filter/1D-variational framework to retrieve temperature, water vapor, and hydrometeor profiles from ground-based microwave radiometers (GMWRs). Designed to overcome classical 1D-Var weaknesses in cloudy conditions, TCKF1D-Var enforces moist-thermodynamic consistency and closes the retrieval with a simple single-moment microphysics (WSM3), linking the state vector to cloud liquid/ice water. Its cost function is reformulated as a dimensionless ratio, removing explicit dependence on climatological

background and observation error covariances and preserving rapidly evolving signals. Key methodological innovations are: (1) using virtual potential temperature ($\theta_v$) as the control variable so temperature, humidity, pressure and hydrometeors adjust jointly under moist-adiabatic constraints; (2) adopting a ratio-based cost function to avoid suppressing transient features by mis-specified B/R matrices; and (3) adding a diagnostic microphysics closure to capture vertical hydrometeor structure when RTTOV-gb radiative transfer alone is insufficient.


A North China deployment of 43 GMWR sites (seven with collocated radiosondes) shows consistent gains. Relative to ERA5 and a classical 1D-Var, TCKF1D-Var substantially reduces systematic biases in temperature and humidity (largest temperature bias reductions above ~2 km; strongest humidity bias reductions from the surface to ~5.5 km). Temperature RMSE is comparable to ERA5 and lower than 1D-Var below ~8.5 km. Humidity RMSE is improved over 1D-Var in the

near-surface layer (0–1.5 km) but is larger than ERA5 aloft—an issue diagnosed below. Hydrometeor validation against collocated EarthCARE cloud liquid water content profiles (±15 min, 15 km; July 2025) finds TCKF1D-Var produces the smallest biases and lowest RMSE, and best reproduces EarthCARE distributions in the 36–136 mg·m$^{-3}$ range. The collocation sample is modest and seasonally limited, so broader validation is needed. In application to eight short-duration extreme precipitation events (WMO ≥10 mm·h$^{-1}$), TCKF1D-Var strengthens precursory $\theta_v$ signals and lengthens first-alert

lead time from ~6–7 h (ERA5) to ~7.5–8 h while retaining the ~4–4.5 h reconfirmation window; 1D-Var often fails to capture robust precursors.

A focused "RMSE deficit" analysis attributes mid-tropospheric humidity degradation to vertical-resolution mismatch and cross-talk between oxygen (temperature-sensitive) and water-vapor channels: misaligned weighting-function peaks permit

temperature uncertainty to leak into humidity retrievals, especially in the ~1.5–4.5 km layer. This highlights an intrinsic limitation of passive microwave profiling and motivates explicit cross-covariance handling or multi-sensor synergy (e.g., lidar, cloud radar).

Limitations include limited EarthCARE collocations, the simplified WSM3 microphysics (fidelity vs. speed tradeoff), and

humidity RMSE aloft. Recommended future work: extend collocations across seasons and regimes, adopt more sophisticated microphysics or Bayesian uncertainty quantification, implement cross-covariance management or sensor fusion, and



incorporate scattering-aware radiative operators and operational assimilation experiments. Overall, TCKF1D-Var embeds thermodynamic conservation and microphysical closure into an efficient variational retrieval, reducing biases, improving hydrometeor realism, and enhancing heavy-rain precursors—promising gains for continuous GMWR profiling and short-range nowcasting.

**Code availability**

The core code of the TCKF1D-Var framework, including the implementation of the optimal estimation algorithm and minimization routines, was developed in Python 3.10.13, available at https://www.python.org/downloads/release/python-31013/ under the Python Software Foundation License (PSFL). The RTTOV-gb v1.0 radiative transfer model can be obtained from https://nwp-saf.eumetsat.int/site/software/rttov-gb/ under the Radiative Transfer for TOVS (RTTOV) License. The TCKF1D-Var framework source code is hosted at https://github.com/smft/TCKF1D-Var.git (Zhang, 2025) under the GNU Affero General Public License (AGPL). To request full access, please contact Dr. Qi Zhang (zhangqi@cnhyc.com) to be added to the user group.

**Data availability**

The ground-based microwave radiometer (GMWR) observations used in this study were collected at Anqing Station and form part of the operational network of the China Meteorological Administration (CMA). These data are publicly available under CMA's data-sharing policy (https://data.cma.cn/). ERA5 reanalysis data are available from the Copernicus Climate Data Store (https://cds.climate.copernicus.eu/) under the License to Use Copernicus Products. Radiosonde data can be accessed via https://www.ncei.noaa.gov/data/igra/ (last accessed on 4 August 2025, under maintenance starting 22 August 2025) under a Creative Commons Attribution 4.0 International License (CC BY 4.0). EarthCARE cloud retrieval product (CPR_CLD_2A) is available from https://earth.esa.int/eogateway/catalog/earthcare-esa-l2-products under CC BY 4.0. The TCKF1D-Var thermodynamic and hydrometeor profile dataset generated in this study is deposited at https://doi.org/10.5281/zenodo.17083973 (Zhang, 2025) under CC BY 4.0.

**Author contribution**

J.P.G. and T.M.C. planned the measurement campaign. T.M.C., M.S., and B.D. conducted the ground-based observations. Q.Z. developed the TCKF1D-Var framework, performed the coding and data analysis, and prepared the manuscript. T.M.C. revised the manuscript. J.P.G., M.S., and B.D. provided funding.



**Competing interests**

The authors declare no conflicts of interests.

**Acknowledgements**

to be added upon acceptance.

**Financial support**

This research is supported by the Ministry of Science and Technology of China under grant 2024YFC3013001, National Nature Science Foundation of China under grant 42307132.

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
