# Peer review of "Retrieving Atmospheric Thermodynamic and Hydrometeor Profiles Using a Thermodynamic-Constrained Kalman Filter 1D-Var Framework Based on Ground-based Microwave Radiometer"

_EGUsphere, 2025_

## Referee Comment (RC1)

**Referee Report — Minor Revision**

Manuscript ID: egusphere-2025-4381
Title: Retrieving Atmospheric Thermodynamic and Hydrometeor Profiles Using a Thermodynamic-Constrained Kalman Filter 1D-Var Framework Based on Ground-based Microwave Radiometer

**Summary statement**

This manuscript presents a novel Thermodynamic-Constrained Kalman Filter 1D-Var (TCKF1D-Var) framework to retrieve atmospheric profiles from ground-based microwave radiometer data. The method introduces a physically consistent approach using virtual potential temperature as the control variable, a ratio-based cost function, and a diagnostic microphysics closure.

The paper is clearly written and technically sound, addressing an important gap in cloudy-condition retrievals. Validation with radiosonde and EarthCARE data convincingly shows improved accuracy over conventional 1D-Var and ERA5. I find the study well-suited for Geoscientific Model Development and recommend minor revision before acceptance.

**General comments**

The manuscript meets GMD's standards for methodological rigor and relevance. The proposed framework is innovative and the results are promising. However, several clarifications and small improvements are necessary to enhance transparency, reproducibility, and interpretability.

The authors should particularly clarify the theoretical justification of the ratio-based cost function, provide more quantitative evidence in the discussion of mid-tropospheric humidity degradation, and improve data/code availability in line with GMD policy.

**Specific comments**

1. Clarification of the ratio-based cost function (Eq. 4)
> The use of a dimensionless, ratio-based cost function is an interesting innovation. Please elaborate on:
>> a) How subtracting unity affects numerical stability and convergence behavior.
>> b) Whether normalization issues arise when observed or simulated brightness temperatures approach zero.
>> c) A brief comparison with the conventional covariance-weighted formulation.

2. Uncertainty quantification and statistical significance
> The performance metrics (bias, RMSE) are presented without uncertainty ranges. Please include standard deviations or confidence intervals, or indicate whether improvements are statistically significant.

3. Microphysics parameterization and coupling
> The coupling between the WSM3 single-moment microphysics scheme and the thermodynamic constraint is not entirely clear. Please expand on how liquid/ice water contents influence the state vector and cost function. A schematic or equation would be helpful.

4. Limited EarthCARE validation sample
> Validation is based only on July 2025 data (around 60 collocated profiles). Please explicitly acknowledge this limitation and discuss whether the conclusions may vary with season or location.

5. Figure readability and accessibility
> The font size in Figures 4–6 is rather small. Please adjust the figure layout so that all symbols, units, and legends are clearly readable and distinguishable.

**Technical corrections**

1. Correct minor grammatical errors (e.g., "profiles shows" → "profiles show"; "biase" → "biases").

2. Maintain consistent notation for virtual potential temperature (θv) in equations and figure captions.

3. Define all acronyms (ERA5, RTTOV-gb, CLWC, WSM3) upon first mention in both abstract and text.

4. Add final acknowledgements before publication.

---

## Referee Comment (RC2)

**Referee Report — Major Revision**

Manuscript ID: egusphere-2025-4381

Title: Retrieving Atmospheric Thermodynamic and Hydrometeor Profiles Using a Thermodynamic-Constrained Kalman Filter 1D-Var Framework Based on Ground-based Microwave Radiometer

**Summary statement:**

This manuscript comes up with a new method, TCKF1D-Var, to retrieve atmospheric profiles from GMWR. This method differs from conventional 1D-Var that employs a ratio-based cost function independent of prescribed background and observation error covariances, and integrate a diagnostic microphysics closure to represent liquid and ice water.

Clear results are shown that lower errors are achieved in cloudy circumstances. However, there are several issues need to be answered before acceptance. I recommend major revision.

**Major Comments:**

1. Line 211: In the figure, it is not clear that "the differences among the three products are mainly confined to the boundary layer." The authors may clarify whether this statement is supported by the figure or revise it accordingly.
   Lines 217–219: In Fig. 5f, the random errors of the TCKF1D-Var temperature profiles remain significantly smaller than those of 1D-Var above 2000 m, which seems inconsistent with the statement "increase substantially" in the text.

2. In Section 4.1.4, the potential cause of the water vapor RMSE deficit is analyzed. As a reviewer not specialized in this field, I would appreciate if the authors could clarify why a similar phenomenon does not occur in the temperature vertical profile?

3. In the overall comparison among TCKF1D-Var, 1D-Var, and ERA5, it appears that temperature differences between TCKF1D-Var and ERA5 are generally small, whereas 1D-Var exhibits large errors in the upper atmosphere, and below 500 m errors remain high. These issues seem unresolved and warrant further discussion. For the first two points, could they be attributed to the dependence on the R and B matrices? If so, it may be slightly unfair to generalize, as in other cases (with different R and B) 1D-Var might perform better. Moreover, since the new cost function depends explicitly on GMWR observations, it is unsurprising that TCKF1D-Var outperforms 1D-Var when the distance between radiosonde and GMWR is minimal. Clarifying these points would be helpful.

4. Only seven sites are equipped with radiosonde observations. Therefore, in composite analysis, large differences may arise between mean bias and RMSE for the same variable. For instance, in Figs. 4c and 4d, the mean bias of water vapor at ~1700 m appears larger than the RMSE, which is mathematically implausible. A similar issue occurs between Figs. 5g and 5h. Please check these results.

5. What criterion is used for the histogram bins in Fig. 9? Are the results sensitive to the

division of water vapor ranges?

**Minor comments:**

1. Lines 75 and 78: The abbreviations should be unified—either GMWR or MWR.

2. In Figure 1, there are 44 stations, which does not match "43" in Line 75.

3. In Section 2.3, it is recommended to include the accuracy information of CPR_CLD_2A.

4. What is the underlying reason for the larger differences between TCKF1D-Var and ERA5 during daytime?

5. Lines 237–239: It seems that Figs. 6i and 6m (rather than 6n) are being analyzed. Moreover, the statement "TCKF1D-Var also exhibits reduced temperature errors below 5 km compared to ERA5 and 1D-Var, while above 5 km its performance is comparable to 1D-Var" corresponds to Fig. 6i, and "ERA5 shows similar errors to 1D-Var below 3 km but becomes less accurate above this level" corresponds to Fig. 6m. Please separate these analyses to avoid confusion.

6. Please ensure that the title and content of Table 1 appear on the same page.

7. Correct the repeated "Figure 8" in the title of Fig. 8.

8. Although Taylor et al. (2007) and Garcia-Carreras et al. (2010) are cited to justify using the temporal moving anomaly of virtual potential temperature as an early-warning indicator, it is recommended to briefly clarify the underlying mechanism.

9. The results in Figs. 10 and 11 are somewhat repetitive. It is recommended to either combine these figures and the corresponding analysis, or present the results without Fig. 11 for conciseness.

---

## Referee Comment (RC3)

I acknowledge the authors' prompt responses to the comments. However, there are still some outstanding questions that merit further discussion.

**Major Comments**:

1.  Line 211: In the figure, it is not clear that "the differences among the three products are mainly confined to the boundary layer." The authors may clarify whether this statement is supported by the figure or revise it accordingly. Lines 217–219: In Fig. 5f, the random errors of the TCKF1D-Var temperature profiles remain significantly smaller than those of 1D-Varabove 2000 m, which seems inconsistent with the statement "increase substantially" in the text.

    **Reply:**

    We sincerely apologize for the confusion caused by our original wording. We agree that the previous statement may have led to a misinterpretation and did not accurately reflect what is shown in the figure. The sentence has been revised to: "For temperature mean bias, the differences between the TCKF1D-Var and the ERA5 are mainly confined to the boundary layer." (Line 242–243)

    **Re:**

    I am wondering whether the interpretation of Fig. 5 is fully consistent with the statement in the manuscript. Specifically, Fig. 5a appears to suggest that the differences between the TCKF1D-Var and ERA5 mainly occur above 600 m, whereas only minor differences are evident within the boundary layer. Similarly, Fig. 5e shows nearly no differences between the TCKF1D-Var and ERA5. These results do not seem to be consistent with the statement that "for temperature mean bias, the differences between the TCKF1D-Var and the ERA5 are mainly confined to the boundary layer." Clarification or further explanation would be helpful.

    **Reply:**

    We thank the reviewer for pointing out the inconsistency between the text and the results in Fig. 5f. We apologize for the misinterpretation created by the earlier description. The corresponding statement has been corrected to: "However, at night(Figure 6f), the random errors of the TCKF1D-Var temperature profiles increase substantially above 8500 m, shifting from being equivalent to ERA5 during daytime to comparable levels." (Line 248–250)

    **Re:**

    In the response, the figure reference may need to be corrected to "Figure 5f" instead of "Figure 6f". Furthermore, the term "comparable levels" is somewhat unclear, and it might be helpful to replace it with a more explicit description (e.g., "slightly higher").

2.  In the overall comparison among TCKF1D-Var, 1D-Var, and ERA5, it appears that temperature differences between TCKF1D-Var and ERA5 are generally small, whereas 1D-Var exhibits large errors in the upper atmosphere, and below 500 m errors remain high. These issues seem unresolved and warrant further discussion. For the first two points, could they be attributed to the dependence on the R and B matrices? If so, it may be slightly unfair to generalize, as in other cases (with different R and B) 1D-Var might perform better. Moreover, since the new cost function depends explicitly on GMWR observations, it is unsurprising that TCKF1D-Var outperforms 1D-Var when the

distance between radiosonde and GMWR is minimal. Clarifying these points would be helpful.

**Reply:**

We deeply appreciate the reviewer for this thoughtful comment and fully agree that the performance of 1D-Var is closely linked to the specification of the background (B)and observation (R) error covariance matrices. We acknowledge that the temperature differences shown in the manuscript—particularly the larger upper-level deviations in 1D-Var and the persistent errors below 500 m—may partially reflect the characteristics of the chosen B and R matrices. Our intention is not to suggest that1D-Var is inherently inferior; indeed, from the standpoint of computational efficiency and operational robustness, 1D-Var offers clear advantages over the proposedTCKF1D-Var method. The central objective of this study is instead to introduce an alternative retrieval approach that satisfies dynamical constraints and incorporates microphysical parameterizations, providing the community with a complementary option beyond 1D-Var. The comparisons included in this work aim to demonstrate that the performance of TCKF1D-Var can reach or exceed that of 1D-Var under the tested conditions.

We also recognize the reviewer's concern regarding regional representativeness. The evaluation sites in northern China do have localized meteorological characteristics, and we do not exclude the possibility that 1D-Var may outperform TCKF1D-Var in other regions — particularly those with weaker water vapor variability or less baroclinicity. While alternative atmospheric profile references exist, radiosonde observations remain the widely accepted standard for directly probing upper-air thermodynamic structure. Thus, using radiosondes as verification data is essential. To mitigate the limitation of sparse co-located radiosonde stations, we additionally compared both retrieval methods against ERA5 reanalysis. The combined results indicate that TCKF1D-Var can effectively exploit GMWR observations to improve the thermodynamic structure relative to ERA5, especially in the lower and middle troposphere.

A synthesized clarification addressing these points has been added to the "Summary and Concluding Remarks" section of the revised manuscript (Line 431-443) to ensure that the discussion is transparent and balanced, reading as: "We also acknowledge that the performance of the classical 1D-Var approach is inherently shaped by the prescribed background (B) and observation (R) error covariance matrices, and the differences highlighted in this study should not be interpreted as a universal limitation. Rather than positioning TCKF1D-Var as a replacement for 1D-Var, our intention is to provide a complementary retrieval framework that incorporates moist-thermodynamic constraints and a microphysical closure, features that are not explicitly represented in the classical formulation. The evaluation sites in North China exhibit regional characteristics, and it is fully plausible that in regimes with weaker humidity gradients or reduced baroclinicity, 1D-Var may perform similarly or even more favorably. Radiosonde observations remain an essential benchmark for upper-air thermodynamic verification, and to address the limited availability of co-located soundings, additional comparisons with ERA5 were included. Overall, the combined evaluation suggests that TCKF1D-Var can extract additional thermodynamic information from GMWR measurements and thus serves as a useful complement to existing 1D-Var techniques

under the conditions examined. These considerations have been incorporated to ensure that the inter-method comparison is presented within a balanced and context-appropriate framework."

**Re:**

Thank you for your detailed reply. However, perhaps due to my previous imprecise wording, I would like to clarify that the relatively large errors below 500 m are not confined to the 1D-Var results. Elevated errors are also evident in ERA5 and TCKF1D-Var, as shown in Figs. 4b–4d, 5a–5d, 5f, and 5h.

In some cases (e.g., Fig. 5e), the error increases upward from near zero, whereas in the other cases listed above, the error decreases from relatively large values near the surface. This contrast represents another interesting feature that merits further explanation.

In addition, my primary concern previously was that the low errors of TCKF1D-Var relative to observations may partly arise from an inherently unfair comparison. Given that TCKF1D-Var does not explicitly incorporate background (B) or observation (R) error covariance matrices and relies strongly on the observations themselves, it is expected that low errors would be obtained if the GMWR observations are sufficiently accurate (e.g., comparable to radiosonde measurements). However, I now realize that this characteristic may instead highlight an **important advantage of TCKF1D-Var**, namely its potential as an effective alternative in regions where GMWR observations are available but radiosonde data are sparse or absent.

3. Only seven sites are equipped with radiosonde observations. Therefore, in composite analysis, large differences may arise between mean bias and RMSE for the same variable. For instance, in Figs. 4c and 4d, the mean bias of water vapor at ~1700 m appears larger than the RMSE, which is mathematically implausible. A similar issue occurs between Figs. 5gand 5h. Please check these results.

**Reply:**

We deeply appreciate the reviewer for pointing out this issue. To address the concern, we have added confidence intervals to Figure 4 (now Figure 5) and Figure 5(now Figure 6) to better illustrate the variability of the statistics. However, we acknowledge that this addition alone does not fully resolve the specific concern regarding the apparent discrepancy between mean bias and RMSE at certain altitudes.

We also recognize that the evaluation was conducted using radiosonde observations from seven sites on 7 July 2025, with two launches per day, yielding a total of 434measurements. While this satisfies the traditional statistical definition of a large sample, it remains insufficient to fully address variations under certain conditions, for example around ~1700 m above ground level, where the retrieval results exhibit relatively large fluctuations. In future work, we plan to increase the sample size for radiosonde verification to improve statistical robustness under such specific conditions. From another perspective, although the TCKF1D-Var results show fluctuations in retrieval accuracy around ~1700 m, the classical 1D-Var maintains relatively stable performance in this layer. This observation is consistent with the reviewer's suggestion that additional attention is required in this altitude range and supports the value of carefully interpreting the variability seen in limited-site composites.

**Re:**

What I pointed out is that in the manuscript the mean bias of water vapor at~1700 m appears larger than the RMSE, which is mathematically implausible given the standard definitions: for errors $e_i$ =model - observation, the root mean square error (RMSE) and mean bias (MB) satisfy $\text{RMSE} = \sqrt{\overline{e^2}}$, $\text{MB} = \overline{e}$, and therefore $\text{RMSE} \geq | \text{MB}|$ (because $\overline{e^2} - \overline{e}^2 = \text{Var}(e) \geq 0$). If the plotted MB exceeds the RMSE, that suggests an inconsistency. Please check and clarify the corresponding analysis.

**Minor Comments:**

1.  The accuracy information for CPR_CLD_2A I recommend is the retrieved hydrometeor profile errors compared to the ground observations such as radiosonde.

---

## Author Comment (AC3)

**Review Comment 1**

Clarification of the ratio-based cost function (Eq. 4)

The use of a dimensionless, ratio-based cost function is an interesting innovation. Please elaborate on:

a) How subtracting unity affects numerical stability and convergence behavior.

b) Whether normalization issues arise when observed or simulated bightness temperatures approach zero.

c) A brief comparison with the conventional covariance-weighted formulation.

**Author Response:**

We thank the reviewer for the insightful comments and interest in our ratio-based cost function. Our detailed responses are as follows:

**a) Effect of subtracting unity on numerical stability and convergence.**

We sincerely thank the reviewer for the valuable comment and for pointing out this important numerical consideration. As described in the manuscript, subtracting unity in the cost function can "enhance the contribution of the initial analysis increment to the cost function and reduce the number of iterations" (lines 155–156). We fully acknowledge that, due to the finite precision of floating-point arithmetic, a potential loss of significant digits may occur, which could affect the numerical stability of the computation. To address this issue, both $\frac{H_{(x)}}{y}$ and $\frac{\theta_{v(x)}}{\theta_{v(x_o)}}$ are scaled to the range of $-1$ to $1$ by multiplying them with an amplification factor calculated from $max(abs(1\left/\frac{H_{(x)}}{y}-1\right.), abs(1\left/\frac{\theta_{v(x)}}{\theta_{v(x_o)}}-1\right.))$, where $max()$ denotes the maximum operator and $abs()$ represents the absolute value operator. Moreover, all input values are converted to double precision prior to initializing the minimization algorithm, in order to further ensure numerical stability and robustness.

For clairity, we have added the discussion below to the manuscript: "Due to the finite precision of floating-point arithmetic, a loss of significant digits may occur,

potentially compromising the numerical stability of the computation. To mitigate this issue, both the observed ($H_{(x)}\big/y$) and simulated ($\theta_{\nu(x)}\big/\theta_{\nu(x_o)}$) variables are normalized to the range of −1 to 1 using an amplification factor derived from

$$max(abs(1\bigg/\left|\frac{H_{(x)}}{y}-1\right|),abs(1\bigg/\left|\frac{\theta_{\nu(x)}}{\theta_{\nu(x_o)}}-1\right|)),$$ where $max()$ and $abs()$ denote the

maximum and absolute value operators, respectively. Furthermore, all input values are converted to double precision before the initialization of the minimization algorithm to enhance numerical stability and robustness" (Lines 161 – 166).

**b) Normalization issues near zero brightness temperature.**

We sincerely thank the reviewer for raising this thoughtful question. Such cases may occasionally occur under clear-sky or non-weather conditions. However, this issue can be effectively mitigated by increasing the number of iterations during the minimization process, which ensures stable convergence. In addition, the normalization treatment, introduced in the revised manuscript (lines 161–166), can further alleviates the potential instability caused by near-zero brightness temperatures. Moreover, the quality control module also alleviates the potential instability by removing the observed brightness temperature whose departure against the simulation is smaller than the noise-equivalent temperature difference.

For clairity, we have added the the discussion below to the manuscript "To avoid normalization issues when observed and simulated brightness temperatures are very close to each other, the quality control module automatically discards GMWR channel observations whose brightness temperature departures from the simulated values are smaller than the noise-equivalent temperature difference" (Lines 166 – 169).

**c) Comparison with the conventional covariance-weighted formulation.**

We appreciate the reviewer's insightful comment regarding this issue. Specifically, the ratio-based cost function minimizes the relative deviation between observed and simulated brightness temperatures rather than their absolute difference. This

approach effectively normalizes the residuals, rendering them dimensionless and ensuring more balanced contributions among different frequency channels. Moreover, the ratio-based formulation is less sensitive to multiplicative calibration or gain errors and better reflects the logarithmic response characteristics of microwave radiative transfer. These advantages have been demonstrated in previous studies and are now explicitly discussed in the revised manuscript.

In the revised manuscript, we have clarified the rationale and advantages of using the ratio-based cost function compared with the conventional difference-based formulation as follows: "It ensures balanced channel weighting, as each channel is normalized by its own magnitude and channels with smaller brightness temperatures are no longer underrepresented during optimization. It also achieves better physical consistency, since the ratio-based form is closer to the logarithmic radiative response of microwave observations, making the inversion more physically meaningful. Finally, it offers enhanced robustness to calibration biases, being less sensitive to multiplicative gain or calibration errors and therefore improving retrieval performance under low signal-to-noise conditions." (Lines 169–175).

**Review Comment 2**

Uncertainty quantification and statistical significance

The performance metrics (bias, RMSE) are presented without uncertainty ranges. Please include standard deviations or confidence intervals, or indicate whether improvements arestatistically significant.

**Author Response:**

We sincerely appreciate the reviewer's valuable comment regarding the quantification of uncertainty and statistical significance. Following this suggestion, confidence intervals have been added to Figures 5, 6, 7, and 9 to better illustrate the variability and robustness of the results. Since the root-mean-square error (RMSE) values already provide a comprehensive measure of the overall deviations, no additional modifications were made to those metrics.

**Review Comment 3**

Microphysics parameterization and coupling:

The coupling between the WSM3 single-moment microphysics scheme and thethermodynamic constraint is not entirely clear. Please expand on how liquid/ice watercontents influence the state vector and cost function. A schematic or equation would behelpful.

**Author Response:**

We thank the reviewer for this constructive comment. In the revised manuscript, we have clarified the coupling between the thermodynamic constraint and the WSM3 single-moment microphysics scheme, and we have added a schematic (now Figure 4) to illustrate the iterative process.

As described in Section 3.2.2, the retrieval begins with the calculation of virtual potential temperature from the priori (background) profiles of pressure, temperature, and water vapor mixing ratio. The cost function minimization then adjusts these thermodynamic variables using the observed GMWR brightness temperatures to generate intermediate profiles. The WSM3 microphysics scheme dynamically updates cloud water and cloud ice mixing ratios based on the intermediate thermodynamic fields and the priori hydrometeor profiles, ensuring thermodynamic and microphysical consistency at each iteration. The updated hydrometeor contents (liquid and ice) subsequently influence the forward-simulated brightness temperatures through the radiative transfer operator, thereby affecting the cost function and its gradient. The iteration continues until the convergence criterion is met, yielding the final analysis fields of pressure, temperature, water vapor, and hydrometeors. The discussions above have been added to the manuscipt (Lines 185 – 194) read as follows: "The coupling between the thermodynamic constraint and the WSM3 single-moment microphysics scheme is illustrated in Figure 4. The procedure begins with the calculation of the virtual potential temperature from the priori thermodynamic and hydrometeor profile. These fields serve as the initial state for the

cost function minimization, where the cost function iteratively adjusts the pressure, temperature, and water vapor mixing ratio using the GMWR brightness temperature observations to produce intermedium profiles. The WSM3 microphysics scheme then dynamically updates the cloud water and cloud ice mixing ratios based on the intermediate pressure, temperature, and water vapor profiles, together with the priori hydrometeor fields. This coupling ensures physical consistency between the thermodynamic state and the microphysical processes during each iteration. If the convergence criterion is satisfied, the resulting profiles of temperature, humidity, and hydrometeors are designated as the final analysis. Otherwise, the updated fields are fed back into the next iteration as new initial conditions until convergence is achieved."

**Review Comment 4**

Limited EarthCARE validation sampleValidation is based only on July 2025 data (around 60 collocated profiles). Please explicitly acknowledge this limitation and discuss whether the conclusions may vary with season or location.

**Author Response:**

We appreciate the reviewer's valuable comment. We acknowledge that the validation dataset, consisting of approximately 60 collocated EarthCARE profiles, is limited in sample size. This constraint indeed restricts the statistical representativeness of the hydrometeor validation results.

However, July was deliberately selected as the test period because the prevailing large-scale circulation over North China during this month frequently produces various types of intense convective systems, including mesoscale convective complexes, squall lines, and stratiform precipitation events. These conditions make July particularly representative of the summer cloud and precipitation regimes in this region, allowing for the evaluation of retrieval performance under diverse hydrometeor conditions.

We also fully recognize that using only July data cannot capture the potential seasonal variability of cloud water and ice characteristics, and thus the current validation results may not fully reflect performance differences across different seasons or locations. We have explicitly discussed this limitation in the revised manuscript and plan to extend the validation to additional months and regions in future work to further assess seasonal dependence.

The discussion from (Lines 425 – 433) read as follows: "This study demonstrates that the TCKF1D-Var framework efficiently integrates thermodynamic constraints and microphysical closure into a unified variational retrieval system, substantially reducing biases, improving hydrometeor profile realism, and enhancing heavy-rain

precursor detection. These results highlight its potential for continuous GMWR profiling and short-range nowcasting applications. However, current validation relies on about 60 collocated EarthCARE profiles from July 2025, which limits the statistical robustness of hydrometeor evaluation and the representativeness of seasonal variability. July was chosen because the prevailing synoptic patterns over North China frequently produce diverse convective systems—making it a suitable test period. Future work will extend evaluations across seasons and regions, employ more advanced microphysics and Bayesian uncertainty quantification, and incorporate multi-sensor fusion and scattering-aware radiative operators to further improve retrieval robustness and operational applicability."

**Review Comment 5**

Figure readability and accessibilityThe font size in Figures 4–6 is rather small. Please adjust the figure layout so that all symbols,units, and legends are clearly readable and distinguishable.

**Author Response:**

We appreciate the reviewer's helpful suggestion. The layouts of Figures 4–6 (now as Figures 5 – 7) have been revised to improve readability. Font sizes for all labels, units, and legends have been enlarged, and the overall figure clarity and color contrast have been enhanced to ensure accessibility and visual consistency throughout the manuscript.

**Reviewer Technical corrections:**

1. Correct minor grammatical errors (e.g., "profiles shows" → "profiles show"; "biase" → "biases").

2. Maintain consistent notation for virtual potential temperature (θv) in equations and figure captions.

3. Define all acronyms (ERA5, RTTOV-gb, CLWC, WSM3) upon first mention in both abstract and text.

4.  Add final acknowledgements before publication.

**Author Response:**

1. The grammatical errors "profiles shows" at Line 214 and "biase" at Line 411 has been corrected.

2. We confirm that the notation for virtual potential temperature is consistant in the revised manuscript.

3. Missing acronyms at Lines 21, 25, 77, and 290 have been added.

4. Acknowledgements have been added.

---

## Author Comment (AC4)

**Summary statement:**

This manuscript comes up with a new method, TCKF1D-Var, to retrieve atmospheric profiles from GMWR. This method differs from conventional 1D-Var that employs a ratio-based cost function independent of prescribed background and observation error covariances, and integrate a diagnostic microphysics closure to represent liquid and ice water.

Clear results are shown that lower errors are achieved in cloudy circumstances. However, there are several issues need to be answered before acceptance. I recommend major revision.

**Reply:**
We sincerely thank the reviewer for the thoughtful and constructive evaluation of our manuscript. We deeply appreciate the reviewer's recognition of the novelty of the proposed TCKF1D-Var method, particularly its ratio-based cost function and the integration of a diagnostic microphysics closure to represent cloud liquid and ice water. We are also grateful for the positive acknowledgement of the improved retrieval performance in cloudy conditions.

We fully agree with the reviewer that several important issues require clarification and further analysis before the manuscript can be considered for acceptance. In the revised version, we have carefully addressed all concerns raised in the detailed comments.

We deeply appreciate the reviewer's recommendation for major revision and have substantially improved the manuscript accordingly. We hope that the revised version satisfactorily addresses all concerns and meets the standards required for publication.

**Major Comments:**

**1.** Line 211: In the figure, it is not clear that "the differences among the three products are mainly confined to the boundary layer." The authors may clarify whether this statement is supported by the figure or revise it accordingly. Lines 217–219: In Fig. 5f, the random errors of the TCKF1D-Var temperature profiles remain significantly smaller than those of 1D-Var above 2000 m, which seems inconsistent with the statement "increase substantially" in the text.

**Reply:**
We sincerely apologize for the confusion caused by our original wording. We agree that the previous statement may have led to a misinterpretation and did not accurately reflect what is shown in the figure. The sentence has been revised to: "For temperature mean bias, the differences between the TCKF1D-Var and the ERA5 are mainly confined to the boundary layer." (**Line 242–243**)

We thank the reviewer for pointing out the inconsistency between the text and the results in Fig. 5f. We apologize for the misinterpretation created by the earlier description. The corresponding statement has been corrected to: "However, at night (Figure 6f), the random errors of the TCKF1D-Var temperature profiles increase substantially above 8500 m, shifting from being equivalent to ERA5 during daytime to comparable levels." (**Line 248–250**)

**2.** In Section 4.1.4, the potential cause of the water vapor RMSE deficit is analyzed. As a reviewer not specialized in this field, I would deeply appreciate if the authors could clarify why a similar phenomenon does not occur in the temperature vertical profile?

**Reply:**
We deeply appreciate the reviewer for this insightful question. We deeply appreciate that, for readers not specialized in microwave radiative transfer or retrieval physics, the asymmetric behavior between humidity RMSE and temperature RMSE may not be immediately intuitive.

To clarify this, we have added an explicit explanation in Section 4.1.4. The key reason a similar RMSE degradation does not occur in the temperature vertical profile is that the temperature retrieval is fundamentally less susceptible to cross-variable leakage for the following reasons: **1.** Temperature channels dominate the information content and have stronger, better-separated weighting functions. The oxygen absorption lines provide well-defined temperature weighting functions that peak at multiple altitudes. These functions overlap less with water-vapor-sensitive channels than vice-versa, which reduces the propagation of humidity-related uncertainties into the temperature retrieval. **2.** Humidity channels have only weak indirect sensitivity to temperature. Although humidity weighting functions are affected by temperature through absorption line broadening, this dependency is substantially weaker compared with the strong temperature sensitivity of oxygen channels. As a result, humidity uncertainties do not significantly contaminate the temperature solution. **3.** Temperature retrieval is more strongly constrained by measurements. The oxygen channels used for temperature have higher signal-to-noise ratios and stronger Jacobians. Their cumulative information content allows the inversion to remain well-conditioned, preventing RMSE amplification in the mid-troposphere. **4.** Humidity retrieval suffers more from vertical-resolution mismatch. The mismatch between humidity and temperature weighting functions (1.5–4.5 km) induces leakage primarily from temperature into humidity, not the other way around. This asymmetry is consistent with prior microwave retrieval theory (Hewison, 2007; Löhnert & Maier, 2012).

To address the reviewer's concern, we have added the following explanation to Section 4.1.4 in the revised manuscript: "A similar RMSE degradation does not occur in the temperature profiles because the oxygen absorption channels provide stronger and more vertically distinct weighting functions, which dominate the temperature information content and are only weakly affected by humidity-related uncertainties. In contrast, water-vapor-sensitive channels exhibit weaker constraints and overlapping sensitivity with temperature weighting functions in the 1.5 – 4.5 km layer, making humidity retrievals more vulnerable to temperature – humidity cross-talk." (**Line 300-306**)

**3.** In the overall comparison among TCKF1D-Var, 1D-Var, and ERA5, it appears that temperature differences between TCKF1D-Var and ERA5 are generally small, whereas 1D-Var exhibits large errors in the upper atmosphere, and below 500 m errors remain high. These issues seem unresolved and warrant further discussion. For the first two points, could they be attributed to the dependence on the R and B matrices? If so, it may be slightly unfair to generalize, as in other cases (with different R and B) 1D-Var might perform better. Moreover, since the new cost function depends explicitly on GMWR observations, it is unsurprising that TCKF1D-Var outperforms 1D-Var when the distance between radiosonde and GMWR is minimal. Clarifying these points would be helpful.

**Reply:**

We deeply appreciate the reviewer for this thoughtful comment and fully agree that the performance of 1D-Var is closely linked to the specification of the background (B) and observation (R) error covariance matrices. We acknowledge that the temperature differences shown in the manuscript—particularly the larger upper-level deviations in 1D-Var and the persistent errors below 500 m—may partially reflect the characteristics of the chosen B and R matrices. Our intention is not to suggest that 1D-Var is inherently inferior; indeed, from the standpoint of computational efficiency and operational robustness, 1D-Var offers clear advantages over the proposed TCKF1D-Var method. The central objective of this study is instead to introduce an alternative retrieval approach that satisfies dynamical constraints and incorporates microphysical parameterizations, providing the community with a complementary option beyond 1D-Var. The comparisons included in this work aim to demonstrate

that the performance of TCKF1D-Var can reach or exceed that of 1D-Var under the tested conditions.

We also recognize the reviewer's concern regarding regional representativeness. The evaluation sites in northern China do have localized meteorological characteristics, and we do not exclude the possibility that 1D-Var may outperform TCKF1D-Var in other regions — particularly those with weaker water vapor variability or less baroclinicity. While alternative atmospheric profile references exist, radiosonde observations remain the widely accepted standard for directly probing upper-air thermodynamic structure. Thus, using radiosondes as verification data is essential. To mitigate the limitation of sparse co-located radiosonde stations, we additionally compared both retrieval methods against ERA5 reanalysis. The combined results indicate that TCKF1D-Var can effectively exploit GMWR observations to improve the thermodynamic structure relative to ERA5, especially in the lower and middle troposphere.

A synthesized clarification addressing these points has been added to the "Summary and Concluding Remarks" section of the revised manuscript (**Line 431-443**) to ensure that the discussion is transparent and balanced, reading as: "We also acknowledge that the performance of the classical 1D-Var approach is inherently shaped by the prescribed background (B) and observation (R) error covariance matrices, and the differences highlighted in this study should not be interpreted as a universal limitation. Rather than positioning TCKF1D-Var as a replacement for 1D-Var, our intention is to provide a complementary retrieval framework that incorporates moist-thermodynamic constraints and a microphysical closure, features that are not explicitly represented in the classical formulation. The evaluation sites in North China exhibit regional characteristics, and it is fully plausible that in regimes with weaker humidity gradients or reduced baroclinicity, 1D-Var may perform similarly or even more favourably. Radiosonde observations remain an essential benchmark for upper-air thermodynamic verification, and to address the limited availability of co-located soundings, additional comparisons with ERA5 were included. Overall, the combined evaluation suggests that TCKF1D-Var can extract additional thermodynamic information from GMWR measurements and thus serves as a useful complement to existing 1D-Var techniques under the conditions examined. These considerations have been incorporated to ensure that the inter-method comparison is presented within a balanced and context-appropriate framework."

**4.** Only seven sites are equipped with radiosonde observations. Therefore, in composite analysis, large differences may arise between mean bias and RMSE for the same variable. For instance, in Figs. 4c and 4d, the mean bias of water vapor at ~1700 m appears larger than the RMSE, which is mathematically implausible. A similar issue occurs between Figs. 5g and 5h. Please check these results.

**Reply:**

We deeply appreciate the reviewer for pointing out this issue. To address the concern, we have added confidence intervals to Figure 4 (now Figure 5) and Figure 5 (now Figure 6) to better illustrate the variability of the statistics. However, we acknowledge that this addition alone does not fully resolve the specific concern regarding the apparent discrepancy between mean bias and RMSE at certain altitudes.

We also recognize that the evaluation was conducted using radiosonde observations from seven sites on 7 July 2025, with two launches per day, yielding a total of 434 measurements. While this satisfies the traditional statistical definition of a large sample, it remains insufficient to fully address variations under certain conditions, for example around ~1700 m above ground level, where the retrieval results exhibit relatively large fluctuations. In future work, we plan to increase the sample size for

radiosonde verification to improve statistical robustness under such specific conditions.

From another perspective, although the TCKF1D-Var results show fluctuations in retrieval accuracy around ~1700 m, the classical 1D-Var maintains relatively stable performance in this layer. This observation is consistent with the reviewer's suggestion that additional attention is required in this altitude range and supports the value of carefully interpreting the variability seen in limited-site composites.

**5.** What criterion is used for the histogram bins in Fig. 9? Are the results sensitive to the division of water vapor ranges?

**Reply:**

We deeply appreciate the reviewer for raising the important question regarding the binning criterion used in Fig. 9. To enhance transparency and ensure that the histogram-based comparison of cloud liquid water content (CLWC) reflects physically meaningful statistical distributions, we followed the studies (Zhang et al., 2021; Mroz et al., 2023) whic have used histogram and probability–density analyses for LWC and related microphysical parameters. Accordingly, for Fig. 9 (now as Figure 10) we grouped the CLWC retrievals and reference observations into a set of bins whose boundaries were chosen to (1) capture the main modes of the LWC/CLWC distribution and (2) align with typical value ranges reported in cloud microphysics literature, ensuring that each bin contains a sufficiently large sample for meaningful comparison. We have added a statement in the revised manuscript to clarify this, for example in the Methods section: "Histogram bins in Figure 10 are defined to ensure sufficient sample counts in each interval for robust frequency comparisons, following established practice in cloud-microphysics statistical analyses (Zhang et al., 2021; Mroz et al., 2023)." (**Line 333–334**)

To the question "Are the results sensitive to the division of water vapor ranges?" Our current study does not explicitly investigate the sensitivity of the results to the division of water-vapor ranges. Based on the limited physical understanding relevant to our present framework, cloud liquid water content is generally positively correlated with the ambient water-vapor abundance, suggesting that some dependence on humidity partitioning may indeed exist. However, a rigorous quantification of this sensitivity lies beyond the scope of the present analysis. We fully agree that this is an interesting and meaningful direction, and we will explore the relationship between environmental water vapor stratification and cloud-water retrieval performance in our future work.

**Reference:**

Mroz, K., Treserras, B. P., Battaglia, A., Kollias, P., Tatarevic, A., and Tridon, F.: Cloud and precipitation microphysical retrievals from the EarthCARE Cloud Profiling Radar: the C-CLD product, Atmos. Meas. Tech., 16, 2865 – 2888, https://doi.org/10.5194/amt-16-2865-2023, 2023.

Zhang, Y., Chen, S., Tan, W., Chen, S., Chen, H., Guo, P., Sun, Z., Hu, R., Xu, Q., Zhang, M., Hao, W., and Bu, Z.: Retrieval of Water Cloud Optical and Microphysical Properties from Combined Multiwavelength Lidar and Radar Data, Remote Sens., 13(21), 4396. https://doi.org/10.3390/rs13214396, 2021.

**Minor comments:**

**1.** Lines 75 and 78: The abbreviations should be unified—either GMWR or MWR.

**Reply:**

We thank the reviewer for pointing this out. We have corrected the inconsistency: the abbreviation has been unified to "GMWR" throughout the manuscript (including the instances at **Line 75–78**). The change is reflected in the revised manuscript and in the tracked-changes file.

**2.** In Figure 1, there are 44 stations, which does not match "43" in Line 75.

**Reply:**

We thank the reviewer for noting this. The mistyping has been corrected, and the abbreviation has now been consistently unified throughout the manuscript.

**3.** In Section 2.3, it is recommended to include the accuracy information of CPR_CLD_2A.

**Reply:**

We have added the accuracy information of the EarthCARE CPR_CLD_2A product in Section 2.3. The revised manuscript now states (**Lines 104–105**): "The active radar observations from EarthCARE provide vertically resolved cloud liquid and ice water content with high sensitivity to optically thick clouds, with a target radar reflectivity accuracy better than 2.7 dB."

**4.** What is the underlying reason for the larger differences between TCKF1D-Var and ERA5 during daytime?

**Reply:**

We thank the reviewer for this insightful question. The manuscript has been revised to clarify the underlying mechanism. In brief, the larger daytime differences between TCKF1D-Var and ERA5 primarily arise from enhanced boundary-layer instability and stronger diurnal variability, which amplify temperature–humidity coupling in passive microwave retrievals. During daytime, solar heating intensifies turbulent mixing and increases the vertical heterogeneity of temperature and water vapor. As a result, TCKF1D-Var — being driven directly by GMWR observations and thermodynamic constraints — responds more strongly to these rapidly evolving features, whereas ERA5 tends to represent smoother background structures due to model diffusion and data-assimilation temporal averaging. This contrast naturally leads to larger daytime departures between the two. Similar daytime–nighttime contrast in vertical thermodynamic heterogeneity and radiometer information content has also been documented in boundary-layer physics and microwave-retrieval studies (Löhnert et al., 2012). Hence, enhanced daytime instability provides a physically consistent explanation for the larger differences between TCKF1D-Var and ERA5 observed in our results.

**Reference:**

Löhnert, U. and Maier, O.: Operational profiling of temperature using ground-based microwave radiometry at Payerne: prospects and challenges, Atmos. Meas. Tech., 5, 1121–1134, https://doi.org/10.5194/amt-5-1121-2012, 2012.

**5.** Lines 237–239: It seems that Figs. 6i and 6m (rather than 6n) are being analyzed. Moreover, the statement "TCKF1D-Var also exhibits reduced temperature errors below 5 km compared to ERA5 and 1D-Var, while above 5 km its performance is comparable to 1D-Var" corresponds to Fig. 6i, and "ERA5 shows similar errors to 1D-Var below 3 km but becomes less accurate above this level" corresponds to Fig. 6m. Please separate these analyses to avoid confusion.

**Reply:**

We fully agree with your comment regarding the mismatch in figure references and the mixing of two separate analyses. Following your suggestion, we have revised the text accordingly and separated the two discussions to avoid ambiguity. The corrections have been implemented in the revised manuscript at Line 269.

**6.** Please ensure that the title and content of Table 1 appear on the same page.

**Reply:**

We have adjusted the layout, and the table title and the table now appear on the same page in the revised manuscript (**Line 320**).

**7.** Correct the repeated "Figure 8" in the title of Fig. 8.

**Reply:**

We deeply appreciate the reviewer for pointing this out. The duplicated Figure 8 has been deleted (**Line 329**).

**8.** Although Taylor et al. (2007) and Garcia-Carreras et al. (2010) are cited to justify using the temporal moving anomaly of virtual potential temperature as an early-warning indicator, it is recommended to briefly clarify the underlying mechanism.

**Reply:**

We deeply appreciate the suggestion. We have clarified the underlying mechanism for using the temporal moving anomaly of virtual potential temperature as an early-warning indicator. In the revised manuscript (**Lines 355–357**), the text now reads: "Following the approach proposed by Taylor et al. (2007) and Garcia-Carreras et al. (2010), we adopt the temporal moving anomaly of virtual potential temperature as an early-warning indicator, which removes slowly varying background signals associated with large-scale processes and diurnal variations."

**9.** The results in Figs. 10 and 11 are somewhat repetitive. It is recommended to either combine these figures and the corresponding analysis, or present the results without Fig. 11 for conciseness.

**Reply:**

We deeply appreciate the suggestion regarding the potential redundancy between Figures 10 and 11. In response, Figure 11 has been moved to Appendix A ("Sensitivity of Virtual Potential Temperature Anomaly to Temporal Averaging Window") and is now labeled as Figure A1.

Correspondingly, the related discussion (lines 348 – 356) has been rewritten for clarity and conciseness. The revised text now reads: "Using the same methodology, we recalculated the time – height evolution of the virtual potential temperature anomaly with a reduced temporal averaging window (Figure A1 in Appendix A), and the gradients of the anomaly variations become weaker compared to those in Figure 11, owing to the shorter averaging window. Nevertheless, both ERA5 and TCKF1D-Var profiles still exhibit the characteristic transition of the anomaly from positive to negative about 7 – 8 h prior to rainfall onset. Although the warm anomaly tongue intrusion remains detectable in both products, its intensity is reduced. When adopting $-0.75$ K as the early-warning threshold, the signal becomes indistinct under the 4.5-hour averaging window, whereas it is enhanced and temporally stabilized within about 2 hours of the precipitation onset when using 6.0-hour and 7.5-hour windows. Consistent with the previous findings, the 1D-Var (Figure A1 c, f, and i) profiles fail to extract effective early-warning signals for heavy rainfall."

---

## Author Comment (AC5)

**Major Comments:**

1. I am wondering whether the interpretation of Fig. 5 is fully consistent with the statement in the manuscript. Specifically, Fig. 5a appears to suggest that the differences between the TCKF1D-Var and ERA5 mainly occur above 600 m, whereas only minor differences are evident within the boundary layer. Similarly, Fig. 5e shows nearly no differences between the TCKF1D-Var and ERA5. These results do not seem to be consistent with the statement that "for temperature mean bias, the differences between the TCKF1D-Var and the ERA5 are mainly confined to the boundary layer." Clarification or further explanation would be helpful.

   **Re:**

   Thank you very much for this insightful comment. We agree that the original wording in the manuscript was not sufficiently clear and may have caused confusion when interpreted together with Fig. 5 (now Fig. 6). Our original intention was to emphasize that, in plain language, for temperature mean bias, the differences between TCKF1D-Var and ERA5 are pretty much limited within (mostly confined to) the boundary layer. To address this issue and improve clarity, we have revised the corresponding sentence in the manuscript. It now reads: "For temperature mean bias, the differences between the TCKF1D-Var and ERA5 are predominantly limited within the boundary layer, while detectable improvements are found above 3000 m above ground level." (**Line 243 – 245**)

2. In the response, the figure reference may need to be corrected to "Figure 5f" instead of "Figure 6f". Furthermore, the term "comparable levels" is somewhat unclear, and it might be helpful to replace it with a more explicit description (e.g., "slightly higher").

   **Re:**

   We thank the reviewer for the careful check and helpful suggestion. Concerning the figure reference, we would like to explain that the figure numbering was updated during the revision process in response to the comments from Reviewer #1.

Specifically, an additional figure entitled "Figure 4. Schematic of the coupling between the thermodynamic constraint and the WSM3 single-moment microphysics scheme" was inserted after the original Figure 3, which led to the renumbering of subsequent figures. As a result, the correct reference in the revised manuscript is Figure 6f.

We also appreciate the suggestion regarding the wording. The term "comparable levels" was indeed not sufficiently clear, and we have revised the sentence accordingly to provide a more explicit description. The revised sentence now reads: "However, at night (Figure 6f), the random errors of the TCKF1D-Var temperature profiles increase substantially above 8500 m, changing from being comparable to ERA5 during daytime to slightly higher than those of ERA5." (**Line 250 – 251**)

3. Thank you for your detailed reply. However, perhaps due to my previous imprecise wording, I would like to clarify that the relatively large errors below 500 m are not confined to the 1D-Var results. Elevated errors are also evident in ERA5 and TCKF1D-Var, as shown in Figs. 4b–4d, 5a–5d, 5f, and 5h. In some cases (e.g., Fig. 5e), the error increases upward from near zero, whereas in the other cases listed above, the error decreases from relatively large values near the surface. This contrast represents another interesting feature that merits further explanation. In addition, my primary concern previously was that the low errors of TCKF1D-Var relative to observations may partly arise from an inherently unfair comparison. Given that TCKF1D-Var does not explicitly incorporate background (B) or observation (R) error covariance matrices and relies strongly on the observations themselves, it is expected that low errors would be obtained if the GMWR observations are sufficiently accurate (e.g., comparable to radiosonde measurements). However, I now realize that this characteristic may instead highlight an important advantage of TCKF1D-Var, namely its potential as an

effective alternative in regions where GMWR observations are available but radiosonde data are sparse or absent.

**Re:**

We deeply appreciate the further clarification and for highlighting this interesting feature. We agree that the relatively large errors below 500 m are not confined to the 1D-Var results, but are also evident in ERA5 and TCKF1D-Var, as shown in the figures you referenced. This behavior reflects a common characteristic of near-surface temperature retrievals rather than a limitation of a specific method.

From the perspective of the GMWR measurement principle, elevated errors in the lowest atmospheric layers are expected. GMWR observations represent vertically integrated brightness temperatures, and the weighting functions of the oxygen and water vapor absorption channels exhibit strong overlap near the surface. As a result, the vertical resolution and information content below several hundred meters are inherently ill-defined. In addition, the diurnal cycle of surface emission, reflection effects, and rapidly varying thermodynamic conditions within the surface layer under different weather conditions further contribute to increased uncertainties. These factors affect all retrieval-based products, not only 1D-Var but also the TCKF1D-Var.

Regarding the contrasting vertical error structures, the decreasing error with height can be attributed to the diminishing surface influence, in cases such as Fig. 5e, where errors increase upward from near-zero values, this behavior likely reflects reduced surface emission and reflection effects at nighttime (12:00 UTC, approximately 20:00 local time). The coexistence of these two patterns therefore results from the combined effects of surface influence, atmospheric stability, and the height-dependent information content of MWR observations.

We agree that, because TCKF1D-Var relies primarily on information from ground-based microwave radiometer observations and does not explicitly prescribe

background (B) or observation (R) error covariance matrices, relatively low errors with respect to independent observations may be obtained when the GMWR measurements are of high quality. At the same time, this aspect may raise concerns regarding the interpretation of the comparison results under certain conditions. Moreover, we acknowledge the point that this characteristic also suggests a potential role for TCKF1D-Var in situations where radiosonde observations are limited or unavailable. In this sense, TCKF1D-Var may serve as a complementary approach rather than a replacement, particularly in observation-sparse regions.

4. What I pointed out is that in the manuscript the mean bias of water vapor at~1700 m appears larger than the RMSE, which is mathematically implausible given the standard definitions: for errors =model - observation, the root mean square error (RMSE) and mean bias (MB) satisfy RMSE = $\sqrt[2]{\overline{e^2}}$, MB = $\overline{e}$ , and therefore RMSE ≥ | MB (because $\overline{e^2}$ – $\overline{e}^2$ = Var($e$) ≥ 0). If the plotted MB exceeds the RMSE, that suggests an inconsistency. Please check and clarify the corresponding analysis.

**Re:**

Thank you very much for pointing out this important issue and for carefully examining the statistical consistency of the results. We agree that, by definition, the root mean square error (RMSE) should always be greater than or equal to the absolute value of the mean bias (|MB|), and we appreciate your attention to this detail.

After carefully re-examining the results, we confirm that the situation you noted around approximately 1700 m does indeed occur. However, this feature is present in the ERA5 reanalysis and the conventional 1D-Var retrievals, rather than in the TCKF1D-Var profiles. The TCKF1D-Var results consistently satisfy the expected statistical relationship between RMSE and MB.

Further analysis of Figure 7 indicates that the cases in which this behavior appears are primarily associated with clear and cloudy conditions for both ERA5 and the 1D-Var retrievals. For ERA5, this behavior can be attributed to limitations related to its relatively coarse horizontal resolution, which affects the representation of non-precipitating clouds and associated water vapor structures in the lower to middle troposphere (e.g., Prange et al., 2023; Virman et al., 2021; McDonald et al., 2025). These limitations can introduce systematic biases that dominate the error statistics at certain altitudes.

For the 1D-Var method, the background error covariance matrix is constructed based on the statistical differences between ERA5 reanalysis and radiosonde observations. Consequently, the aforementioned deficiencies of ERA5 are implicitly propagated into the 1D-Var retrievals through the background constraint, leading to similar bias characteristics around 1700 m.

In contrast, no such inconsistency is observed in the TCKF1D-Var profiles. This indicates that the use of a thermodynamic constraint based on virtual potential temperature conservation, together with the coupling to a cloud microphysical parameterization scheme, provides a robust framework for water vapor retrievals under both clear and cloudy conditions.

**Reference:**

Prange, M., Buehler, S. A., and Brath, M.: How adequately are elevated moist layers represented in reanalysis and satellite observations? Atmospheric Chemistry and Physics, 23, 725–741, https://doi.org/10.5194/acp-23-725-2023, 2023.

Virman, M., Bister, M., Räisänen, J., Sinclair, V. A., & Järvinen, H.: Radiosonde comparison of ERA5 and ERA-Interim reanalysis datasets over tropical oceans. Tellus A: Dynamic Meteorology and Oceanography, 73(1), 1929754, https://doi.org/10.1080/16000870.2021.1929752, 2021.

McDonald, A. J., Kuma, P., Panell, M., Petterson, O. K. L., Plank, G. E., Rozliaiani, M. A. H., & Whitehead, L. E.: Evaluating cloud properties at Scott Base: Comparing ceilometer observations with ERA5, JRA55, and MERRA2 reanalyses using an instrument simulator. Journal of Geophysical Research: Atmospheres, 130, e2024JD041754, https://doi.org/10.1029/2024JD041754, 2025.

**Minor Comments:**

1. The accuracy information for CPR_CLD_2A I recommend is the retrieved hydrometeor profile errors compared to the ground observations such as radiosonde.

   **Re:**

   Thank you for this valuable suggestion. We fully agree that independent ground-based observations would be highly desirable for evaluating the accuracy of the retrieved hydrometeor profiles in CPR_CLD_2A.

   We initially attempted to validate the retrievals using routine ground-based observations. However, conventional radiosonde measurements do not provide observations of cloud liquid water content or cloud ice water content, which prevents a direct quantitative comparison with the retrieved hydrometeor mass concentration profiles. Although Ka-band millimeter-wave cloud radars are deployed at some sites, these instruments in our study region currently provide primarily basic reflectivity measurements and do not routinely produce quantitative cloud liquid or ice water content products. This limitation makes a robust and consistent validation against ground-based observations challenging.

   Therefore, we acknowledge that the specific validation requested cannot be fully addressed within the scope of the present study. We have clarified this limitation in the revised manuscript. In future work, we plan to mitigate this shortcoming by

extending the temporal coverage of the dataset and incorporating observations from a larger number of sites, as well as by exploring additional synergistic ground-based measurements where available.

---

## Referee Report (RR1)

**General Comments:**

The authors have provided a thorough, clear, and well-organized response to all comments raised during the first round of review. The revised manuscript has been substantially improved, and the changes made appropriately address the reviewers' concerns. In particular, the additional explanations and revisions have enhanced the clarity, methodological transparency, and overall quality of the study.

After carefully examining the revised version, I find that only one minor issue remains that requires clarification or small revision, as detailed below. This issue does not affect the main conclusions of the paper and can be readily addressed without further analysis.

Therefore, I recommend acceptance after minor revision.

**Specific Comments:**

Regardless of the data source, when the same set of observations is used as reference, the RMSE is generally expected to be larger than the corresponding mean bias (MB), since RMSE incorporates both systematic and random errors. This should hold consistently for the proposed method, the baseline method, and the reanalysis products when evaluated against the same observations.

However, the results presented in the revised manuscript appear to violate this basic relationship in some cases. I therefore suggest that the authors carefully re-examine the calculation and presentation of the RMSE and MB, including the definitions, units, and averaging procedures, to ensure consistency and correctness.

---

## Author Response (AR2)

Dear Editor,

Thank you very much for your message and for giving us the opportunity to further revise our manuscript. We have carefully addressed the remaining minor comments from the reviewer as well as the note provided by the editorial office, as detailed below.

Regarding the map shown in Figure 1, we confirm that it falls under case (b), i.e., the map was created by us based on a background layer reused from another originator. Specifically, the basemap used for Figure 1 is the publicly available "World Imagery" basemap provided by ArcGIS. In accordance with your guidance, we have added a corresponding clarification in the manuscript text (Lines 85–86) and included an explicit credit to the ArcGIS "World Imagery" basemap in the reference section (Lines 524–526).

With respect to the comment raised by Reviewer 2, we have provided a detailed response in the point-by-point reply to the reviewer. In addition, we have revised the manuscript accordingly to address this concern, with the relevant changes implemented in Lines 449–451.

We sincerely appreciate the editor's and reviewers' time and constructive feedback, which have helped us further improve the clarity and quality of the manuscript. We hope that the current revision adequately addresses all remaining concerns.

Kind regards,
Qi Zhang and all co-authors

**Reply to Reivewer 2**

**General Comments:**

The authors have provided a thorough, clear, and well-organized response to all comments raised during the first round of review. The revised manuscript has been substantially improved, and the changes made appropriately address the reviewers' concerns. In particular, the additional explanations and revisions have enhanced the clarity,methodological transparency, and overall quality of the study. After carefully examining the revised version, I find that only one minor issue remains thatrequires clarification or small revision, as detailed below. This issue does not affect the main conclusions of the paper and can be readily addressed without further analysis. Therefore, I recommend acceptance after minor revision.

> **Reply:**
>
> We sincerely thank the reviewer for the careful re-evaluation of our revised manuscript and for the very positive and encouraging comments. We are pleased to hear that the revisions have substantially improved the clarity, methodological transparency, and overall quality of the study, and that our responses have adequately addressed the concerns raised during the first round of review.
>
> We also appreciate the reviewer for pointing out the remaining minor issue. As detailed in the response below, we have carefully addressed this point through clarification and a small revision in the manuscript. We believe that this revision further improves the presentation of the work without affecting the main conclusions.
>
> Thank you again for the constructive feedback and the recommendation for acceptance after minor revision.

**Specific Comments:**

Regardless of the data source, when the same set of observations is used as reference, the RMSE is generally expected to be larger than the corresponding mean bias (MB), since RMSE incorporates both systematic and random errors. This should hold consistently for the proposed method, the baseline method, and the reanalysis products when evaluated against the same observations. However, the results presented in the revised manuscript appear to violate this basic relationship in some cases. I therefore suggest that the authors carefully re-examine the calculation and presentation of the RMSE and MB, including the definitions, units, andaveraging procedures, to ensure consistency and correctness.

> **Reply:**
>
> We fully agree with the reviewer's comment that, when evaluated against the same reference observations, the RMSE is generally expected to be larger than the

corresponding mean bias (MB), as RMSE reflects the combined effects of systematic and random errors.

To further investigate this issue, we conducted an additional evaluation using independent observations from the Beijing station (WMO ID: 54511) for the period from May to July 2025, applying exactly the same retrieval and validation procedures as in the original analysis. The results show that, for both the retrieved profiles and ERA5, the mean bias is consistently smaller than the corresponding RMSE, in agreement with the expected error relationship. This additional test suggests that, although the original experiments included observations from more than 40 stations, the statistics based on one-month data at individual stations may still be subject to sampling limitations. As can be seen from Figure 1 below (panel b), the MB of analysis (red line) and ERA5 (cyan line) at 1700 meter above ground level (purple dashed line) stays around 0.55 g/kg and 0.70 g/kg, while the RMSE (pannel e) at the same height is 0.95 g/kg for both the analysis and the ERA5.

In response to this comment, we have made targeted revisions in the manuscript (**Lines 449 – 451**) of the manuscript to improve clarity and consistency, reading as: "July was selected as the initial test period because the prevailing synoptic conditions over North China frequently give rise to a wide range of convective systems, providing a favorable environment for evaluation; nevertheless, extending the experimental period remains necessary to ensure more robust and statistically representative results".

[Figure]

Figure 1. Validation of temperature, water vapor, and pressure profiles retrieved by TCKF1D-Var (red), ERA5 a priori (cyan) against radiosonde observations for Beijing Station (WMO ID: 54511). The shaded areas denote the 95% confidence intervals that have passed the significance test.

---

## Author Response (AR3)

Dear Editor,

Thank you very much for your kind message and for accepting our manuscript for publication. We sincerely appreciate your support and guidance throughout the review and revision process.

Following the suggestion from the editorial team, we have added the statement "Powered by Esri" to the caption of Figure 1, which is now included in Line 86 of the revised manuscript.

We would like to express our heartfelt thanks for your great assistance during the peer-review and revision stages. It has been a pleasure working with you.

With best regards,
Qi Zhang
On behalf of all authors